# An optimal transformation method for inferring ocean tracer sources and sinks

Jan D. Zika[1,2,3,*] and Taimoor Sohail[1,2,*]

[1]School of Mathematics and Statistics, University of New South Wales, Sydney, Australia
[2]Australian Center for Excellence in Antarctic Science, University of New South Wales, Sydney, Australia
[3]UNSW Data Science Hub (uDASH), University of New South Wales, Sydney, Australia
[*]These authors contributed equally to this work.

**Correspondence:** Jan D. Zika (j.zika@unsw.edu.au)

**Abstract.** The geography of changes in the fluxes of heat, carbon, fresh water and other tracers at the sea surface are highly uncertain and are critical to our understanding of climate change and its impacts. We present a state estimation framework wherein prior estimates of boundary fluxes can be adjusted to make them consistent with the evolving ocean state. In this framework, we define a discrete set of ocean water masses distinguished by their geographical and thermodynamic/chemical properties for specific time periods. Ocean circulation then moves these water masses in geographic space. In phase space, geographically adjacent water masses are able to mix together, representing a convergence, and air-sea property fluxes move the water masses over time. We define an optimisation problem whose solution is constrained by the physically permissible bounds of changes in ocean circulation, air-sea fluxes and mixing. As a proof of concept implementation, we use data from a historical numerical climate model simulation with a closed heat and salinity budget. An inverse model solution is found for the evolution of temperature and salinity consistent with 'true' air-sea heat and fresh water fluxes which are introduced as model priors. When biases are introduced to the prior fluxes, the inverse model finds a solution closer to the true fluxes. This framework, which we call the Optimal Transformation Method, represents a modular, relatively computationally cost effective, open source and transparent state estimation tool that complements existing approaches.

## 1 Introduction

As the climate warms, the ocean acts as a giant reservoir, absorbing excess heat (Cheng et al., 2022) and exchanging vast amounts of biologically critical gasses (Friedlingstein et al., 2022). Accurately projecting future climate change hinges on a deeper understanding of this exchange of properties at the sea surface, and the subsequent ocean response via mixing and circulation. Estimates of past changes in air-sea exchange have large uncertainties, hampering efforts to accurately model them. There is broad disagreement between individual atmospheric reanalysis products on the trends in air-sea heat fluxes since the 1970s, particularly outside the equatorial Pacific (Cheng et al., 2022; Friedlingstein et al., 2022; Chaudhuri et al., 2013; Bentamy et al., 2017), and these trends in air-sea heat fluxes do not correspond with in-situ observations of the change in ocean temperatures over the same period (e.g., Valdivieso et al. (2017)). The same is true for air-sea freshwater flux products, which can deviate from one another and from observations of ocean salinity change significantly (Grist et al., 2016). Therefore,

new techniques are needed to translate observations of the changes in distribution of ocean properties into estimates of the rates
of air-sea exchange, mixing and circulation.

Changes in the concentration of key oceanic properties such as temperature, salinity, oxygen and carbon can be directly
measured. From these observations, air-sea fluxes can be inferred by fitting a physical model of the ocean. This is called
'inverse modelling' or 'state estimation' (Wunsch, 2006). A number of common approaches have been employed in the past to
produce oceanic state estimates, including hindcasts, Four Dimensional Variational Assimilation (4DVAR), Green's Functions
and water mass based methods.

Hindcasts are derived by taking a forward marching numerical model of the ocean, which is initialised with our best guess
of the initial distribution of ocean properties, and forced at the sea surface by observational estimates of the atmospheric
state, including wind, speeds, air, temperature, and humidity. This yields a physically consistent estimate of the state of the
ocean over a given time. With careful consideration of model drift, hindcasts have been used to produce accurate descriptions
(or 'state estimates') of recent ocean temperature changes, and therefore heat fluxes from hindcasts have been interpreted as
providing plausible descriptions of recent changes (Drijfhout et al., 2014; Huguenin et al., 2022). However, such hindcasts do
not typically describe other tracers such as salinity accurately without surface salinity restoring (Griffies et al., 2009).

Four Dimensional Variational Assimilation (Wunsch and Heimbach, 2007, 4DVar, also described as the "adjoint method") is
a more sophisticated extension to hindcasts where, during a model run, the state of the model is differentiated with respect to
initial and boundary conditions. Through iteration, boundary and initial conditions are adjusted (in effect systematically tuned)
to minimise the least squares difference between the model and observations, leading to as physically consistent a model state
as is feasible from which plausible air-sea fluxes result. 4DVar is, however, computationally expensive, meaning simulations
typically focus on the very recent past. For instance, the latest data product from the Estimating the Circulation and Climate of
the Ocean (ECCO) project covers the period 1992-2017 (Forget et al., 2015). In addition, the state estimate is closely tied to
the specific numerical schemes of the model used. For example, if the model's resolution and advection scheme cannot capture
a boundary current accurately, then no change to model boundary and initial conditions can change that.

The state estimation approach we propose here is not intended to be a competitor to 4DVar but rather an alternative approach
with distinct use cases. The method we propose is rooted in both ocean transport and water mass theory, both of which we will
review briefly in the context of state estimation.
A common approach to ocean state estimation, particularly in terms of of ocean tracers, is to consider every point in the
ocean at time $t$, as being a mixture of contributions transported from other regions of the ocean at previous times given by a
'Green's Function' (GF; Haine and Hall, 2002). In its pure form the GF provides a complete description of all aspects of ocean
circulation and mixing, a complete GF is too high dimensional to be solved for using an inverse model (a GF linking each point
in space and time to each other point in space and time and would be eight-dimensional). That said, GF-based methods have
been put to practical use by assuming ocean circulation is steady, and by considering only the connection between a limited
number of surface patches and interior ocean points (Khatiwala et al., 2009; Zanna et al., 2019).

In practice, a GF is inversely fit to a set of observational estimates of both surface and interior concentrations or by directly
calculating the GF-based on a steady numerical model. An adjacent approach is to directly fit a so called 'transport matrix'

(Khatiwala, 2007). GF and transport matrix methods have been used to infer transient changes in the air-sea fluxes of properties (such as anthropogenic carbon; Mikaloff Fletcher et al., 2006; Khatiwala et al., 2009), as well as to infer long-term changes in ocean properties (such as ocean heat content; Zanna et al., 2019; Newsom et al., 2020). In addition to steady state assumptions, implicit in these approaches is the assumption that the air-sea exchange of properties is proportional to the anomaly of that property at the sea surface. These assumptions can lead to substantial errors and restrict the range of variables that can be described (Wu and Gregory, 2022). We aim to develop a method that does not rely on these assumptions.

A water mass is typically defined as a body of water with distinct thermodynamic and/or chemical properties. Water mass based methods are rooted in the fact that *only* sources and sinks of properties at the sea surface and mixing can change the underlying volumetric distribution of water masses in terms of their properties (Groeskamp et al., 2019). For instance, adiabatic ocean circulation cannot directly change the volume of water that is warmer than a given value. Because sources and sinks of properties and mixing are typically far larger near the sea surface than the deep ocean, the properties of water masses are often thought to indicate a common formation history.

Traditional box inverse methods (Wunsch, 1978) and their extensions (such as the tracer contour method Zika et al., 2009) effectively use a water mass approach since properties are conserved within isopycnal layers or along temperature/salinity iso-contours on isopycnals. More recently, the unique properties of water mass transformation have been exploited with the thermohaline inverse method (THIM). In THIM, Groeskamp et al. (2014b) frames the inverse problem in terms of the global conservation of volume in multiple tracer (temperature and salinity) coordinates. This approach has been extended to a regional context with the Regional Thermohaline Inverse Method (Mackay et al., 2018). However, these methods have not been focused on inferring air-sea exchanges (in those examples, air-sea fluxes are taken as known boundary conditions) nor investigating long term changes.

Water mass based methods have been used in a number of studies focused on understanding variability, for example the seasonal cycle of water masses (Groeskamp et al., 2014a; Evans et al., 2014), interannual variability in the North Atlantic (Evans et al., 2017; Josey et al., 2009), long term changes in salinity (Zika et al., 2015a; Skliris et al., 2016) and temperature (Sohail et al., 2021) and the ocean's properties (Sohail et al., 2022; Zika et al., 2021). Here, we will build on these studies and incorporate aspects of Green's Functions-based methods to develop a general, yet relatively simple and intuitive water mass based state estimation tool for the changing ocean, termed the Optimal Transformation Method (OTM) .

In Section 2 we build up the OTM state estimation framework in the most general terms. In Section 3 we discuss a specific implementation of OTM and test this implementation using numerical model data. In Section 4 we present the state estimates and sensitivity tests. In Section 5 we discuss the utility of the framework and conclude.

## 2   Optimal Transformation Method

### 2.1   Prelude

Consider a fluid with a set of conservative tracers $\mathbf{C} = [A, B, ...]^T$, where $A(\mathbf{x}, t)$ is a scalar describing the concentration of the first tracer in space ($\mathbf{x}$) and time ($t$), $B(\mathbf{x}, t)$ the concentration of the second and so on. By "conservative", we mean that, in the

absence of explicit sources and sinks of tracer substance, a parcel of fluid following fluid motion will retain its concentration unless it is irreversibly mixed with other fluid parcels. Furthermore, when a fluid parcel of mass $m_1$ with concentration $\mathbf{C}_1$ mixes with a fluid parcel of mass $m_2$ with concentration $\mathbf{C}_2$, the resulting fluid parcel has mass $m = m_1 + m_2$ and tracer

concentration

$$\mathbf{C}_{mix} = \frac{m_1 \mathbf{C}_1 + m_2 \mathbf{C}_2}{m_1 + m_2}. \tag{1}$$

For the case of only one tracer variable, any fluid parcel with concentration $C_{mix}$ can be formed from a linear combination of 2 other fluid parcels with concentrations $C_1$ and $C_2$ so long as $C_1 \leq C_{mix} \leq C_2$.

We now consider a description of many water masses and many tracers. We define an *early* set of water masses describing

an early period of time being converted into a *late* set of water masses some period of time $\Delta t$ later. Let there be a set of $N$ early water masses with tracer concentrations $\{\mathbf{C}_{0,1}, \mathbf{C}_{0,2}, ..., \mathbf{C}_{0,N}\}$ and $N$ late water masses with $\{\mathbf{C}_{1,1}, \mathbf{C}_{1,2}, ..., \mathbf{C}_{1,N}\}$. In both cases the first subscript denotes the point in time (early = 0; late = 1) and the second denotes the index of the water mass corresponding to that state. To make the mathematics as simple as possible in this Section, each water mass has the same mass, $m$, in the early and late states. We will relax this constraint in the practical implementation of the method (Section 3.3).

If the system is closed, the late water masses are constituted from the early water masses. That is, there is some 'transport' matrix, whose entries $g_{ij}$ represent the mass fraction from the $i$th early water mass used to create the $j$th late water mass. Applying mass conservation we have

$$1 = \sum_{i=1}^{N} g_{ij} \text{ and } 1 = \sum_{j=1}^{N} g_{ij}. \tag{2}$$

In Zika et al. (2021), we solved for $g_{ij}$. by minimizing the amount of warming and cooling water masses had to undergo,

in a root mean square sense, to achieve the observed change in water mass distribution in temperature and salinity coordinates (see also Evans et al. (2014)). Using that approach we are not able to make use of observational estimates of air sea heat and fresh water fluxes nor were we able to impose physics based constraints on mixing driven transformations.

Here we present a method where the influence of sources and sinks of tracer, circulation and mixing are considered separately, which we call the Optimal Transformation Method (OTM). We now discuss how mixing and tracer sources and sinks

can drive transformation and modify the water mass distribution in tracer space.

## 2.2 Mixing driven transformation

Equation (1) describes a situation where two water masses are mixed to form another water mass. More generally, late water masses can be made from a range of fractional contributions from the early water masses. If changes in tracer properties were solely due to fluid mixing, the tracer concentrations of the late water masses would be the mass weighted mean of the early.

That is,

$$\mathbf{C}_{1,j} = \sum_{i=1}^{N} g_{ij} \mathbf{C}_{0,i}. \tag{3}$$

The idea that the properties of the interior ocean water masses are linear combinations of the properties of surface or boundary water masses was exploited by Tomczak (1981) and subsequent authors such as Gebbie and Huybers (2010) to describe the origins of oceanographic water masses. Unlike traditional water mass analysis, which considers the formation of interior water masses from boundary water masses in steady state, we consider the formation of new water masses from old water masses over time and the influence that sources and sinks of tracer at the sea surface have on that transformation.

## 2.3 Sources and sinks of tracer

The ocean is not a closed system. Heat and tracer substances are exchanged at the sea surface and interior sources and sinks of tracer exist due to a range of biological, chemical and physical processes. We will now incorporate such sources and sinks.

The fraction of our $i$th early water mass which is transported to the $j$th late water mass can be subjected to a source or sink of tracer on its route from one to the other. We represent this source as an implied change in tracer concentrations $\mathbf{Q}_{ij}$ along the Lagrangian path taken by the fraction of water $g_{ij}$. That is, the fraction of water ($g_{ij}$) that leaves the early water mass $i$ with tracer concentration $\mathbf{C}_{0,j}$ can be thought of as having been changed to concentration $\mathbf{C}_{0,j} + \mathbf{Q}_{ij}$ by the time it arrives at late water mass $j$.

The late water mass $j$ is formed from the mixture of all the fractions of $g_{ij}$ modified along their respective paths such that its tracer concentration is

$$\mathbf{C}_{1,j} = \sum_{i=1}^{N} g_{ij} \left( \mathbf{C}_{0,i} + \mathbf{Q}_{ij} \right). \tag{4}$$

This provides a complete description of water mass change: the late water masses ($\mathbf{C}_{1,j}$) are formed as the linear combination of fractions ($g_{ij}$) of the early water masses ($\mathbf{C}_{0,i}$), each modified on route by sources and sinks ($\mathbf{Q}_{ij}$).

If we knew the transport and sources/sinks we could use (4) to predict the late state given the early state as a forward problem. In our case, however, we will frame an inverse problem where we have imperfect knowledge of some of the terms in (4).

## 2.4 Solving for the transport matrix and source/sink adjustments

In practise, we do not know any of the 4 terms in (4) with certainty for any tracers in the ocean. We can, however, frame (4) as an inverse problem, and adjust the terms within it to find solutions under certain constraints. Many different strategies could be employed depending on the confidence of the user in the different terms and constraints. We will develop and implement one approach we consider relevant to understanding recent multi-decadal changes in ocean temperature and salinity.

For heat and salt, we consider there to be relatively good confidence in observational estimates of $\mathbf{C}_{1,j}$ and $\mathbf{C}_{0,i}$, poorer confidence in estimates of $\mathbf{Q}_{ij}$ and poor to no confidence in estimates of $g_{ij}$. The concentrations $\mathbf{C}_{1,j}$ and $\mathbf{C}_{0,i}$ can be derived

from ocean temperature and salinity analyses (e.g. (Good et al., 2013)). These come with substantial uncertainties (Cheng et al., 2022; Stammer et al., 2021), but have the benefit of essentially being mappings of directly observed quantities. The source/sink term $\mathbf{Q}_{ij}$ can be inferred from air-sea flux products but these come with larger uncertainties. For example, heat content changes derived from temperature analyses vary by order $0.1\text{W/m}^2$ (e.g. $0.05\text{W/m}^2$ for 1958-2019, Cheng et al., 2022) while those derived from accumulated air-sea heat fluxes typically have biases of order $1\text{W/m}^2$ (e.g. $4\text{W/m}^2$ for 1993-2009, Valdivieso et al., 2017). Finally, we know of no direct way of deriving $g_ij$ from observations. Indeed, $g_ij$ could be derived from a data constrained numerical model, but that would imply it is indirectly derived from the same data used for $\mathbf{C}_{1,j}$, $\mathbf{C}_{0,i}$ and $\mathbf{Q}_{ij}$. We thus consider it reasonable to frame an inverse problem where $\mathbf{C}_{1,j}$ and $\mathbf{C}_{0,i}$ are considered 'known', priors for $\mathbf{Q}_{ij}$ are provided and $g_{ij}$ is merely constrained to obey laws of physics.

We separate the sources and sinks of tracers into a 'prior' estimate and an 'adjustment' such that $\mathbf{Q}_{ij} = \mathbf{Q}_{ij}^{prior} + \mathbf{Q}_{ij}^{adjust}$ and (4) becomes

$$\mathbf{C}_{1,j} = \sum_{i=1}^{N} g_{ij} \left( \mathbf{C}_{0,i} + \mathbf{Q}_{ij}^{prior} \right) + \sum_{i=1}^{N} g_{ij} \mathbf{Q}_{ij}^{adjust}. \tag{5}$$

We aim to derive a solution for $g_{ij}$ such that $\mathbf{Q}_{ij}$ is as 'close' as possible to $\mathbf{Q}_{ij}^{prior}$ (i.e., the air-sea flux adjustment, $\mathbf{Q}_{ij}^{adjust}$ is as small as possible). We therefore use the following cost function:

$$[\text{Cost}] = \sum_{j=1}^{N} \left\| \mathbf{w}_j \left( \sum_{i=1}^{N} g_{ij} \left( \mathbf{C}_{0,i} + \mathbf{Q}_{ij}^{prior} \right) - \mathbf{C}_{1,j} \right) \right\|^2, \tag{6}$$

where $\mathbf{w}_j$ is a relevant weighting (see Section 2.5). The minimisation of the cost (6) combined with constraints (2) and (4) is an inverse problem (hereafter 'the inverse problem'), or more specifically, a linear program for which $g_{ij}$ can be solved for using constrained linear optimisation tools.

Physically, solving for $g_{ij}$ using (6) implies we modify the early water masses with the prior source/sink estimates, then find the geographical rearrangement and mixing of those modified water masses that gets us as close as possible to the later water masses.

Solving for $g_{ij}$ then leads to an estimate of the total source/sink of tracer experienced in transit to the late water mass $j$ via

$$\sum_{i=1}^{N} g_{ij} \mathbf{Q}_{ij}^{adjust} = \mathbf{C}_{1,j} - \sum_{i=1}^{N} g_{ij} \left( \mathbf{C}_{0,i} + \mathbf{Q}_{ij}^{adjust} \right). \tag{7}$$

The accumulated tracer source following the fluid motion from early water mass $i$ to late water mass $j$ is then $\mathbf{Q}_{ij}^{prior} + \mathbf{Q}_{ij}^{adjust}$.

To recapitulate, we have described a method where we find the optimal transport matrix $g_{ij}$ using (6), and then, from this, we find the adjustment required to tracer sources and sinks using (7). We call this an Optimal Transformation Method (OTM) since we are looking for the optimal way in which the waters can be transformed to describe the evolving ocean state given our physical constraints.

OTM is similar to a range of previous water mass based inverse analyses such as (Evans et al., 2014; Groeskamp et al., 2014b; Mackay et al., 2018) in that they attempt to solve for a transformation rate, given existing data for the late and early water masses and tracer sources and sinks.

In Section (3) we discuss the specific practical considerations of our data inputs, the definition of weights ($\mathbf{w}_j$) and the numerical solution. First though, we discuss some general considerations of the choice of weights and additional constraints.

## 2.5 Consideration of weights

Solving (6) without the weight function ($\mathbf{w}_j = 1$) would yield a cost function whereby sources and sinks within all water mass are penalised equally, regardless of their geographical location.

The purpose of $\mathbf{w}_j$ is to favour solutions where the source and sink adjustments are more likely. One case where this is apparent is for tracers with little or no interior source or sink such as conservative temperature (essentially a tracer of heat), salinity (a tracer of fresh water) and anthropogenic tracers such as chlorofluorocarbons. For such tracers, it makes sense to not allow (or at least heavily penalise) fluxes into tracer sources in water masses that do not outcrop. More-over, if the flux of tracer per unit area at the sea surface had a known uncertainty, this could be used to derive the weights as the product of the uncertainty per unit area and the area. This way, adjustments to the tracer sources would incur a higher costs in (6) for water masses that have a small outcrop and/or have low uncertainties in the fluxes over that outcrop. In our toy examples and our application to data from a climate model, we will consider only the case where the uncertainty in the fluxes are the same in a per unit area sense so that the weights are proportional to the inverse of the area.

Furthermore, the weight $\mathbf{w}_j$ can be different for different properties. It is sensible for $\mathbf{w}_j$ to take into account the relative effect of $\mathbf{Q}_{ij}^{adjust}$ on different properties in the cost function. For instance, the user may want to penalise a source of salt, which leads to a 1g/kg change in salinity more than a source of heat leading to a 1K change in temperature.

## 2.6 Additional constraints

We have so far discussed the general case where $N$ early water masses are transformed into $N$ late water masses. Since $g_{ij}$ can be nonzero for all $i$ and $j$, water can be transported from any water mass on the globe to any other. Since some of these transports will be implausible, it is appropriate to place constraints and/or costs on certain parts of the transport matrix, $g_{ij}$.

Here, a range of options are possible, for example a 'speed limit' could be defined permitting water to only travel a certain maximum distance over the time period $\Delta t$. More sophisticated connectivity constraints could be imposed based on vertical and horizontal and/or isopycnal and diapycnal excursions and integrated constraints could be imposed based on energetic considerations. The inverse method described is flexible and allows for such additional constraints to be readily added.

## 2.7 Toy examples

To help explain and develop an intuition for how the Optimal Transformation Method works and is solved, here we discuss a number of toy examples. To make the examples as simple as possible, while still allowing for a range of behaviour, only 3

water masses with two conservative tracers: salinity, $S$ (in grams per kilogram) and temperature, $T$ (in degrees Celsius) are considered.

The toy examples below are illustrated in figures 1 (for examples 1 and 2) and 2 (for examples 3,4 and 5).

### 2.7.1   Example 1: Pure mixing

When there is no prior information given regarding the sources and sinks of tracer ($\mathbf{Q}_{ij}^{prior} = 0$), optimisation of the inverse problem is achieved first by mixing water masses together, then an adjustment is applied to complete the picture.

Three water masses form a triangle in $T - S$ space , initially with $\mathbf{C}_{0,1} = [0,34.6]$, $\mathbf{C}_{0,2} = [4,35]$, $\mathbf{C}_{0,3} = [0,35.4]$ and at a
later time with $\mathbf{C}_{1,1} = [1,34.9]$, $\mathbf{C}_{1,2}=[2,35]$, $\mathbf{C}_{1,3} = [1,35.1]$. In this case the triangle contracts over time to form a smaller triangle. Equations (4) and (2) are satisfied for $\mathbf{Q}_{ij} = \mathbf{0}$ with $g_{ij} = 0.5$ when $i = j$ and $g_{ij} = 0.25$ otherwise. Here the triangle is contracted by mixing the water masses together.

### 2.7.2   Example 2: Pure sources and sinks

Now consider the case where $\mathbf{C}_{0,1} = [1,34.9]$, $\mathbf{C}_{0,2} = [2,35]$, $\mathbf{C}_{0,3} = [1,35.1]$ and $\mathbf{C}_{1,1} = [0,34.6]$, $\mathbf{C}_{1,2} = [4,35]$, $\mathbf{C}_{1,3} = [0,35.4]$.
Here, the triangle expands. Intuitively this cannot be achieved by mixing, which is a convergent process in $T - S$ space. Indeed (4) could be satisfied with $\mathbf{Q}_{ij} = \mathbf{0}$ but only by violating (2) (effectively the water masses would need to be 'unmixed'). With $\mathbf{Q}_{ij}^{prior} = 0$, a minimum cost (6) is found with $g_{ij} = 1$ when $i = j$ and $g_{ij} = 0$ otherwise. So, the change in water masses is achieved not by mixing the water masses, but instead by translating the corners of the triangle outward via adjustment to the sources and sinks ($\sum_{i=1}^{N} g_{ij} \mathbf{Q}_{ij}^{adjust}$).

### 2.7.3   Example 3: Sources and mixing

Consider now an example where the three initial water masses do not change between the early and late periods with $\mathbf{C}_{0,1} = [1,34.9] = \mathbf{C}_{1,1}$, $\mathbf{C}_{0,2} = [2,35] = \mathbf{C}_{1,2}$, $\mathbf{C}_{0,3} = [1,35.1] = \mathbf{C}_{1,3}$. In this case the triangle appears not to move. Now consider prior sources/sinks such that $\mathbf{C}_{0,1} + \mathbf{Q}_{1j}^{prior}=[0,34.6]$, $\mathbf{C}_{0,2} + \mathbf{Q}_{2j}^{prior}=[4,35]$, $\mathbf{C}_{0,3} + \mathbf{Q}_{3j}^{prior}=[0,35.4]$ for all $j$. A solution then exists with no cost, according to (6). That is, a valid solution can be found with mixing alone. This occurs when $g_{ij} = 0.5$ for
$i = j$ and $g_{ij} = 0.25$ otherwise (as in the pure mixing case). In this solution, the sources and sinks expand the triangle, and according to the transport matrix, the water masses are then mixed together, contracting the triangle to achieve an unchanged water mass distribution.

### 2.7.4   Example 4: Sources, mixing and thermohaline circulation

Consider once again a situation where the three initial water masses are the same for the early and late periods with $\mathbf{C}_{0,1} = [1,34.9]$, $\mathbf{C}_{0,2} = [2,35]$, $\mathbf{C}_{0,3} = [1,35.1]$ and $\mathbf{C}_{1,1} = [1,34.9]$, $\mathbf{C}_{1,2} = [2,35]$, $\mathbf{C}_{1,3} = [1,35.1]$. Now consider a prior source/sink such that $\mathbf{C}_{0,1} + \mathbf{Q}_{1j}^{prior} = [0,35.4]$, $\mathbf{C}_{0,2} + \mathbf{Q}_{2j}^{prior} = [0,34.6]$, $\mathbf{C}_{0,3} + \mathbf{Q}_{3j}^{prior} = [4,35]$ for all $j$. Again a solution exists with no cost (6). However, rather than a symmetric matrix we have $g_{12} = 0.5$, $g_{23} = 0.5$, $g_{31} = 0.5$ and $g_{ij} = 0.25$ otherwise. Here

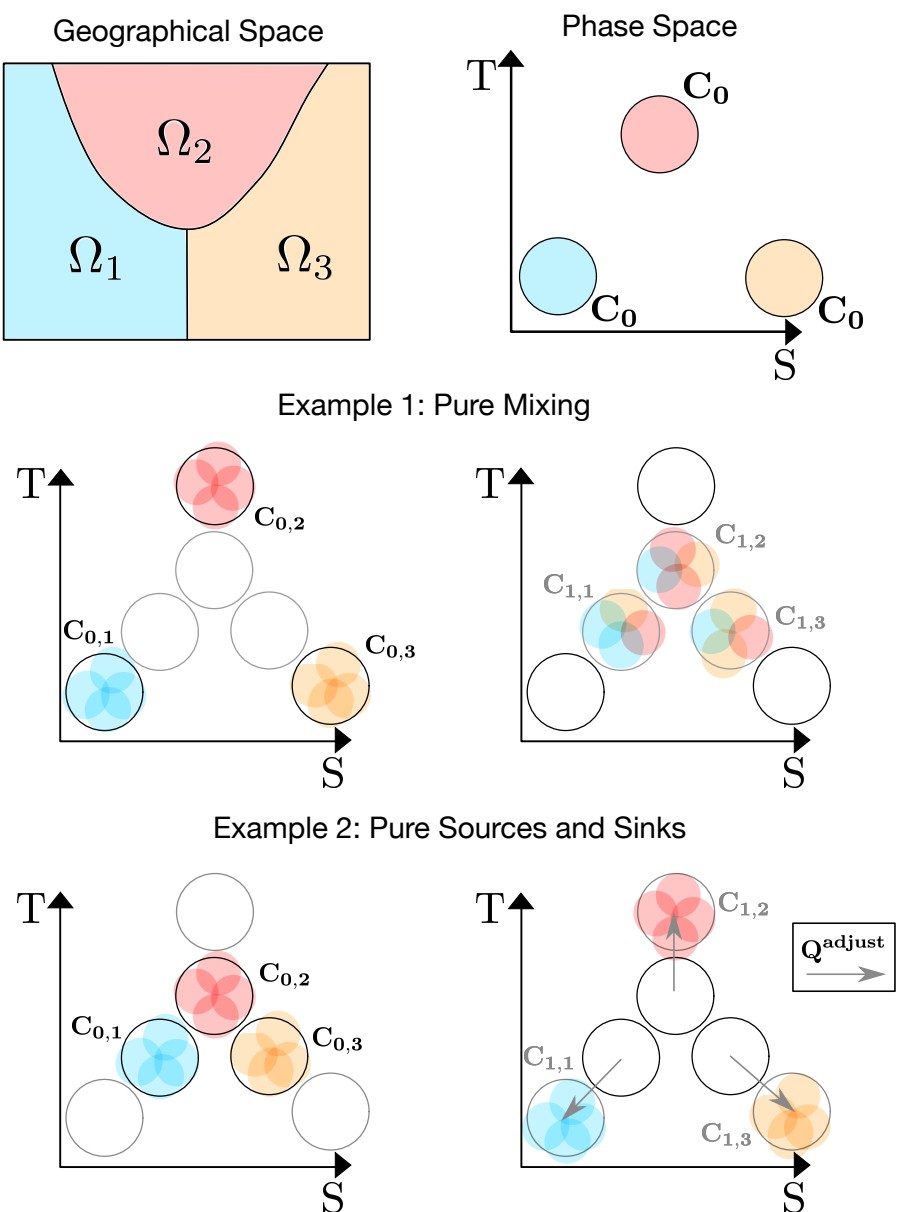

**Figure 1.** Illustration of the method using toy examples with 3 early ($\mathbf{C}_{0,i}$) and 3 late ($\mathbf{C}_{1,j}$) water masses in $T - S$ coordinates. The water masses occupy geographical regions given by $\Omega_{0,i}$. The fraction of the $i$th early water mass that arrives in the $j$th late water masses ($g_{ij}$) is represented by the coloured circles, each representing 1/4 of the water mass it came from and 1/12 of the total mass in the system. For example, in the pure mixing example, 2 blue circles from early water mass 1 (i.e. half of water mass 1) arrive in late water mass 1 so that $g_{11} = 0.5$, while 1 blue circle from early water mass 1 arrives at late water mass 2 so that $g_{12} = 0.25$. Movements in $T - S$ space induced by sources and sinks are shown as arrows (black: priors, $\mathbf{Q}_{ij}^{prior}$; grey: adjustments, $\mathbf{Q}_{ij}^{adjust}$).

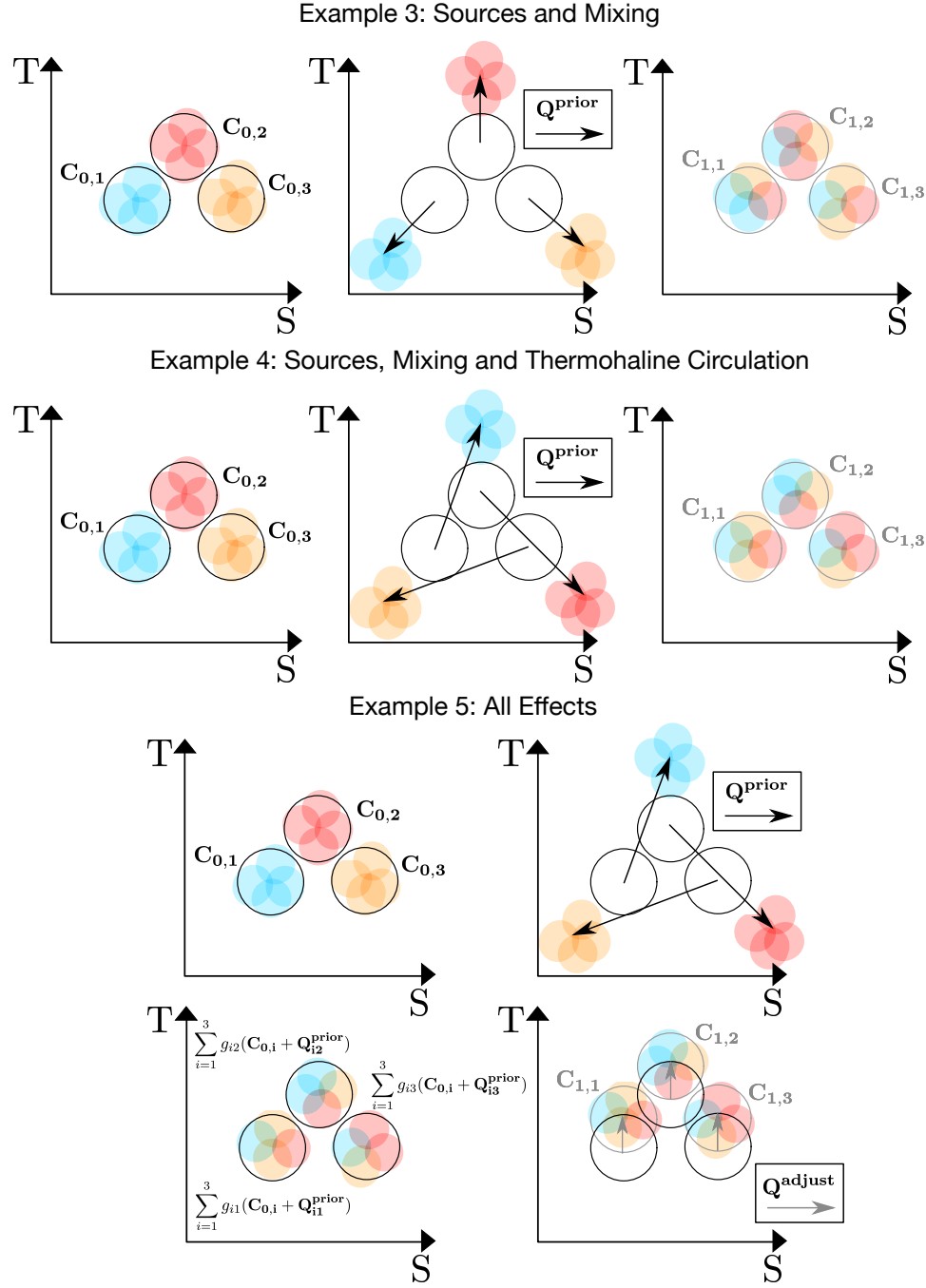

**Figure 2.** As in Figure 1 but for the remaining toy examples.

the transport matrix describes *both* a mixing and a clockwise circulation of the water masses in $T - S$ space. The latter circulation aspect is represented by the anti-symmetric part of the transport matrix. If the water masses are associated with fixed geographical regions, the anti-symmetric part of the transport matrix represents the thermohaline component of the geographical circulation (Zika et al., 2012).

### 2.7.5 Example 5: All effects

Finally, consider the case where the water masses are changing in time with $\mathbf{C}_{0,1}$ = [1,34.9], $\mathbf{C}_{0,2}$ = [2,35], $\mathbf{C}_{0,3}$ = [1,35.1] and $\mathbf{C}_{1,1}$ = [2,34.9], $\mathbf{C}_{1,2}$ = [3,35], $\mathbf{C}_{1,3}$ = [2,35.1]. Let us assume prior sources/sinks, which describe a steady source vs mixing cycle as in the previous example, but do not capture the overall warming, i.e., $\mathbf{C}_{0,1} + \mathbf{Q}_{1j}^{prior}$ = [0,35.4], $\mathbf{C}_{0,2} + \mathbf{Q}_{2j}^{prior}$ = [0,34.6], $\mathbf{C}_{0,3} + \mathbf{Q}_{3j}^{prior}$ = [4,35] for all $j$. In this case no solution exists without a cost (6). With the weights constant, the lowest cost is achieved by the same transport matrix as in the sources, mixing and circulation example, with $g_{12} = 0.5$, $g_{23} = 0.5$, $g_{31} = 0.5$ and $g_{ij} = 0.25$ otherwise. The remaining adjustment to each water mass ($\mathbf{Q}_{ij}^{adjust}$) is then simply [0,1] for all $i$ and $j$. That is, the sources and sinks will satisfy (4) if 1°C of warming is added to each water mass. In this example, different weights could lead to differing distributions of the warming across the water masses and consequent changes in the transport matrix.

## 2.8 Summary of the Optimal Transformation Method

In this section we have outlined a water mass based state estimation framework, the Optimal Transformation Method. OTM relates knowledge of changing ocean tracer distributions to transient ocean transport and mixing. We propose an inverse method based on this framework to infer minimal adjustments to prior estimates of tracer sources and sinks.

In the following sections we will discuss one practical implementation of OTM and assess it using data from a historical climate model simulation.

## 3 Data and implementation

## 3.1 Synthetic data from a historical climate simulation

In Section 2, a general implementation of OTM was presented for any set of tracers. In this work, we demonstrate an implementation of this framework by analysing changes in temperature and salinity (and their associated surface fluxes of heat and freshwater) in a climate model.

We analyse ocean conservative temperature (hereafter temperature or $T$) and ocean practical salinity (hereafter salinity or $S$) from a historical simulation of the ACCESS-CM2 climate model, which forms part of the Australian submission to the 6th generation Climate Model Intercomparison Project (CMIP6). The Modular Ocean Model (MOM, version 5.1) is used as the ocean component of the coupled ACCESS-CM2 model. We analyse the three-dimensional, monthly-averaged conservative temperature and practical salinity field from January 1979 to December 2014 (inclusive) in ACCESS-CM2. Surface fluxes,

$\mathbf{Q_i}$, are obtained from the surface heat and freshwater flux variables, (*hfds* and *wfo* respectively), except in section 4.3, where a reanalysis product is used instead (see below). Surface flux tendencies are obtained by time-integrating the relevant flux variables over the period of interest, then taking a time-derivative over this period, following Sohail et al. (2021, 2022). The early period covers the time period from January 1979 to December 1987, and the late period covers the time period from January 2006 to December 2014, inclusive.

Temperature and salinity exhibit a long-term climate drift in ACCESS-CM2 (further explored by Irving et al. (2020)). Despite this long-term drift, the heat and freshwater budgets close in the model (that is, the globally-integrated cumulative surface flux is equal to the ocean heat and freshwater content change). Provided the heat and freshwater budgets close, the long-term drift in the ACCESS-CM2 model is immaterial for the purposes of validating the OTM state estimation framework laid out in Section 2. Thus, we analyse the drifting historical simulation in this work. Further details on the model spin-up, forcing and drift are provided by Bi et al. (2020); Mackallah et al. (2022); Irving et al. (2020).

To test the performance of the OTM algorithm to a spatially heterogeneous bias, we apply 'known' air-sea fluxes, $\mathbf{Q}_{ij}^{prior}$, from an air-sea reanalysis product, ERA5, in section 4.3 (Hersbach et al., 2020). Produced by the European Centre for Medium-Range Weather Forecasts (ECMWF), ERA5 combines a wide range of atmospheric and oceanic observational products with an operational weather forecasting model (the Integrated Forecasting System (IFS), Cy41r2) using 4D-Var Data Assimilation. The result is a well-constrained, long-term representation of our best estimate of 'known' air-sea fluxes. We assess the two-dimensional, gridded monthly-averaged net surface heat and freshwater fluxes in ERA5 from January 1979 to December 2014. The ERA5 surface flux fields are re-binned onto the ACCESS-CM2 native grid prior to assessment with OTM to ensure the ERA5 global net surface fluxes are accurately captured in the analysis.

## 3.2   Definition of discrete water masses using *Binary Space Partitioning*

The global ocean's temperature-salinity $(T - S)$ distribution is an integrated measure of its hydrographic properties, displaying the volume or mass of the ocean with a characteristic temperature and salinity range (figure 3).

Our OTM state estimation framework considers the transformation from a set of 'early' water masses to a set of 'late' water masses in tracer and geographical space. We split the ocean into 9 basins (following Zika et al., 2021) - the polar North Atlantic, subtropical North Atlantic, equatorial Atlantic, South Atlantic, Indian, South Pacific, Equatorial Pacific, North Pacific and Southern Ocean. Only transport between adjacent ocean basins is permitted in the optimization problem, such that $g_{ij} = 0$ between water masses in non-adjacent basins. Ideally, the discrete representation should be as fine as possible so as to best describe our $T - S$ distribution (i.e., as many discrete water masses as possible), while also considering the distributions representative of different geographical regions. However computational constraints limit the resolution and number of regions possible. Here, we define the discrete water masses using *Binary Space Partitioning* (BSP), following Sohail et al. (2023).

The BSP algorithm recursively sub-divides the mass-weighted $T - S$ distributions along the T- and S-axes $n$ times, resulting in $2^n$ bins which all contain exactly the same mass. BSP represents an improvement over the quadtree coarsening algorithm (as used by Zika et al., 2021) as it results in a predetermined number of bins which hold exactly the same volume. Note that the BSP coarsening presented here is a two-dimensional equivalent to the 1-dimensional tracer-percentile framework introduced by

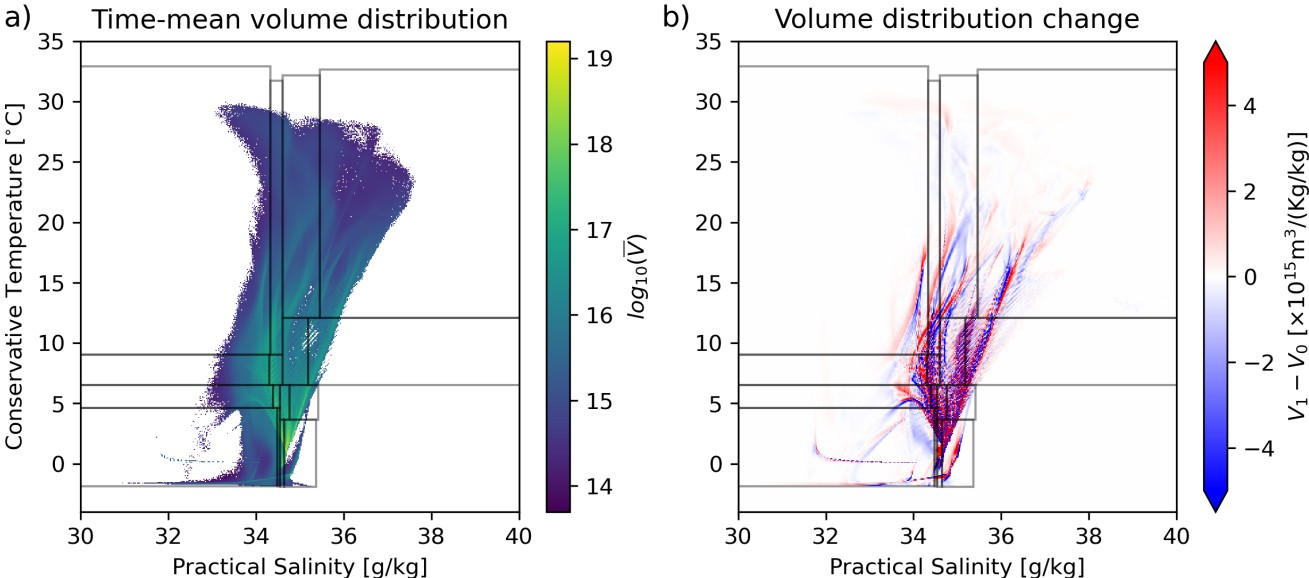

**Figure 3.** (a) The global distribution of ocean volume in $T-S$ space, averaged between January 1979 and December 2014, in the historical simulation of the ACCESS-CM2 climate model. (b) Volume distribution change between the time-averaged 'early' and 'late' periods, defined as January 1979 – December 1987 and January 2006 – December 2014 inclusive, respectively. Black boxes show the 16 bins defined using binary space partitioning, each of which contains 1/16th of the volume of the upper 2000m of the ocean (since these data come from a Boussinesq ocean model, mass and volume are proportional).

Sohail et al. (2021, 2022). Further information on Binary Space Partitioning and its applications in oceanography is provided in Sohail et al. (2023).

### 3.3 Implementation of the inverse model

We recursively subdivide the $T-S$ distribution of the top 2000m of the global ocean in ACCESS-CM2 4 times to yield $2^4 = 16$ classifications of equal volume/mass globally (since ACCESS-CM2's ocean component is Boussinesq, volume and mass are proportional to one another). We further partition these 16 $T-S$ classifications into each of the 9 basins defined above over the full ocean depth. This produces what we define as our 144 'early' and 144 'late' water masses. Each water mass has different tracer concentrations: ($\mathbf{C}_{0,i} = [T_{0,i}, S_{0,i}]$ and $\mathbf{C}_{1,j} = [T_{1,j}, S_{1,j}]$), and due to the splitting by region, a different mass ($m_{1,i}$ and

$m_{0,i}$). Figure 4 shows the mean temperature and salinity of each of these water masses (white dots), as well as the volume (colour) and $T$-$S$ ranges (rectangles) in each basin.

     Each water mass has a corresponding 'mask', $\Omega_i(x,y,z,t)$ defining its geographical location with time ($\Omega_i = 1$ within the water mass and $\Omega_i = 0$ outside; $x$, $y$ and $z$ are latitude, longitude and depth respectively). The outcrop area of water mass $i$ at time $t$ is then, $\iint \Omega_i(x,y,0,t)dA$ and $A_i$ is the time average of that area (defined below).

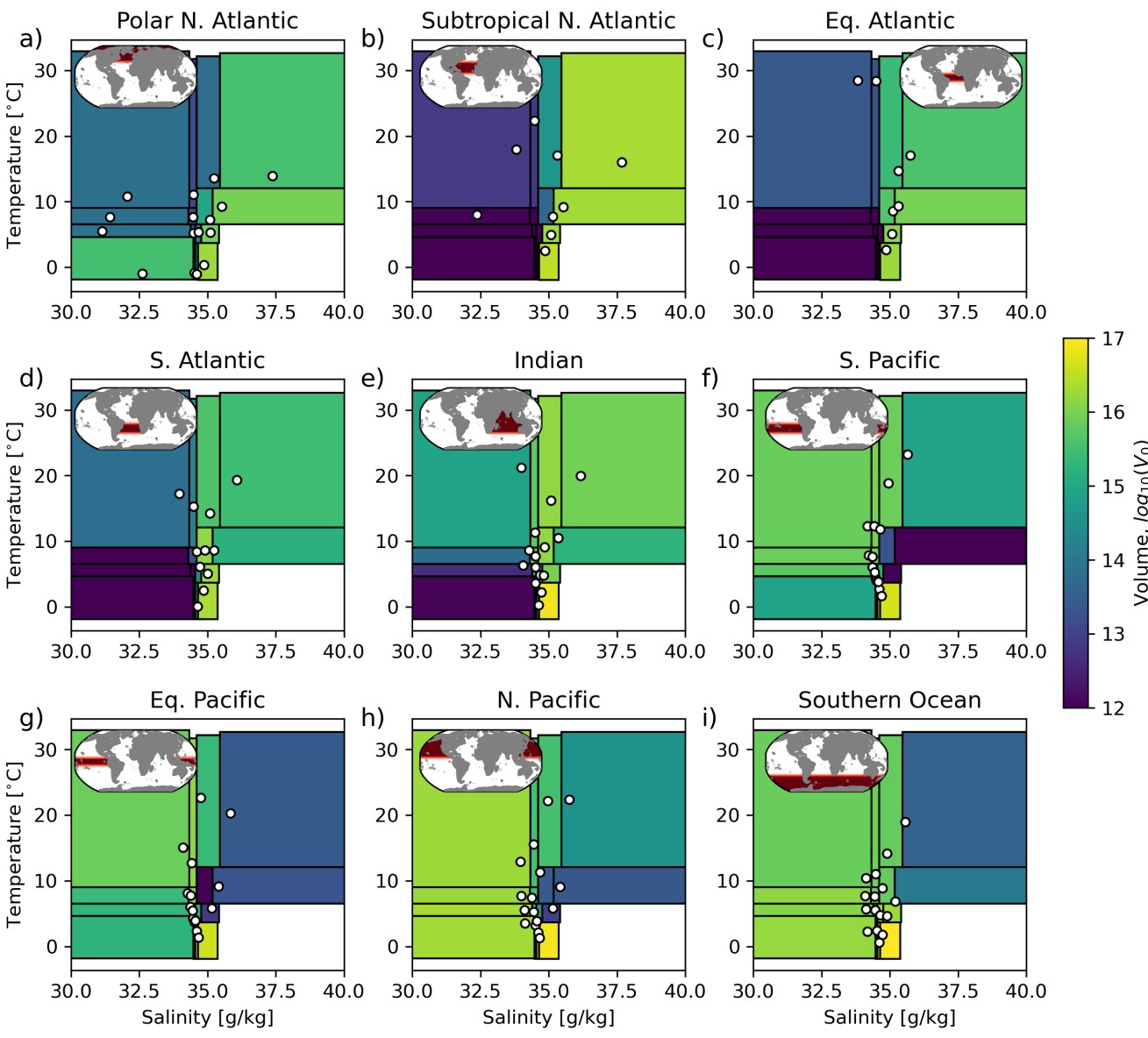

**Figure 4.** Volume (colours) and mean $T$ and $S$ of each of the 16 bins in each of the 9 basins analysed in the 'early' period. Each rectangle represents the range of $T$-$S$ values covered by one of the water masses. The colour of the rectangle represents the volume of water in that bin in that basin. Each white point contained within a rectangle is located at the average $T$-$S$ value of the water in that bin in that basin.

The following hard constraints are placed on the entries of the transport matrix $g_{ij}$ (note variable early and late masses $m_{0,i}$ and $m_{1,i}$ have been incorporated into the constraints below):

$$0 \leq g_{ij} \leq 1; \tag{8}$$

$$m_{1,j} = \sum_{i=1}^{N} m_{0,i} g_{ij}; \tag{9}$$

$$m_{0,i} = \sum_{j=1}^{N} m_{1,j} g_{ij}; \tag{10}$$

$$\mathbf{C}_{1,j} m_{1,j} = \sum_{i=1}^{N} \mathbf{C}_{0,i} m_{0,i} g_{ij} \text{ where } A_j = 0; \tag{11}$$

$$g_{ij} = 0 \text{ if } \Omega_i \text{ and } \Omega_j \text{ are not in the same or adjacent regions.} \tag{12}$$

The above enforce mass conservation (8-10), tracer conservation away from the surface boundary (11) and the inability of water to move further than the adjacent basin (12).

A transport matrix $g_{ij}$ is then sought which minimises the following cost function:

$$[\text{Cost}] = \sum_{j=1}^{N} \left\| \mathbf{w}_j \left( \sum_{i=1}^{N} m_{0,i} g_{ij} \left( \mathbf{C}_{0,i} + \mathbf{Q}_{ij}^{prior} \right) - m_{1,j} \mathbf{C}_{1,j} \right) \right\|^2 \tag{13}$$

with

$$\mathbf{w}_j = \frac{1}{A_j} \left[ \frac{1}{std(T)}, \frac{1}{std(S)} \right]. \tag{14}$$

Effectively, $\mathbf{w}_j$ leads (13) to search for the smallest residual source/sink per unit outcrop area and normalises the impact of temperature and salinity on the residuals relative to their global standard deviations. The additional constraint on $g_{ij}$ (11) ensures that changes to water masses that do not outcrop are achieved purely by redistribution and mixing. In one of the cases we will discuss below (where $\mathbf{Q}_i^{prior} = 0$), our optimiser does not find a feasible solution with this constraint when $A_i = 0$ for some $i$ values. In that case, we set a floor on those areas as the minimum non-zero $A_j$ found for all $j$. This was, in that specific case, the most permissive area constraint we could justify for the problem.

We set the 'prior' change in tracer concentration driven by tracer sources and sinks to the same value for all early water masses $i$ regardless of their path to the late water masses $j$ (so $\mathbf{Q}_{ij}^{prior}$ becomes $\mathbf{Q}_i^{prior}$). We calculate this by integrating the 'known' model air-sea fluxes over the outcrop region of the early water mass and over the time interval between the early and late periods such that:

$$\mathbf{Q}_i^{prior} = \frac{1}{m_{0,i}(t_1 - t_0)} \int_{t_0}^{t_1} \iint \Omega_i(x,y,0,t) \mathbf{q}(x,y,t) dx dy dt \tag{15}$$

where $t_0$ and $t_1$ are mid points of the early and late periods. Above $\mathbf{q}(x,y,t) = [\text{hfds}(x,y,t), -S_0\text{wfo}(x,y,t)] + bias$, where $bias$ is a bias we will introduce in some cases to see what effect incorrect air-sea flux data has on the inverse solution. The above time integral is from the midpoint of the early period to the midpoint of the late period since it is related to the change in average water mass properties between the two periods. (Integrating from the start of the early to the end of the later period would overestimate the sources and sinks.)

In our implementation of the optimisation (13) we aim to minimise the average adjustment to the tracer sources and sinks in a per unit area sense. For this reason we calculate the average outcrop area using the same integral limits as the sources and sinks such that

$$A_i = \frac{1}{t_1 - t_0} \int\limits_{t_0}^{t_1} \iint \Omega_i(x,y,0,t)dxdydt. \tag{16}$$

Equations (8) to (13) define a conic linear optimisation problem. We solve this numerically with the Python based *cvxpy* package, specifying the 'MOSEK' optimisation solver with default settings to obtain a transport matrix $g_{ij}$ which satisfies the constraints described over the time period of interest.

## 4 Results

When a solution for $g_{ij}$ is found by minimising (13), an adjustment to the tracer sources and sinks is implied in order to close the tracer budgets. We diagnose this adjustment via:

$$\mathbf{Q}_j^{adjust} = \mathbf{C}_{1,j} - \frac{1}{m_{1,j}} \sum_{i=1}^{N} m_{0,i} g_{ij} \left( \mathbf{C}_{0,i} + \mathbf{Q}_i^{prior} \right). \tag{17}$$

Once the early water masses have been redistributed and mixed by $g_{ij}$, $\mathbf{Q}_j^{adjust}$ is the remaining change in tracer concentrations required for these mixtures to match the late water mass concentrations, $\mathbf{C}_{1,j}$. We do not attribute different adjustments to the different fractions of the early water masses that make up the late water masses, so that $\mathbf{Q}_j^{adjust}$ is the same for all $i$.

The 'inverse solution' describing the evolution of ocean water masses is then the transport matrix $g_{ij}$ and the implied total sources and sinks of tracer given by $\mathbf{Q}^{prior} + \mathbf{Q}^{adjust}$. Since, in the case of heat and salt, we attribute the sources and sinks to fluxes at the sea-surface, the adjustment term is converted into a flux per unit area and mapped onto geographical coordinates via:

$$\mathbf{q}_{adjust}(x,y,t) = \sum_{j=1}^{N} \frac{m_j}{A_j(t_1 - t_0)} \mathbf{Q}_j^{adjust} \Omega_j(x,y,0,t). \tag{18}$$

Above, the tracer source required to change water mass $j$ by $\mathbf{Q}_j^{adjust}$ is applied as a flux of tracer per unit area. Because of we only infer one adjustment flux per water mass, we are not able to infer more detailed variations in the flux over the spatial extent of the water mass outcrop.

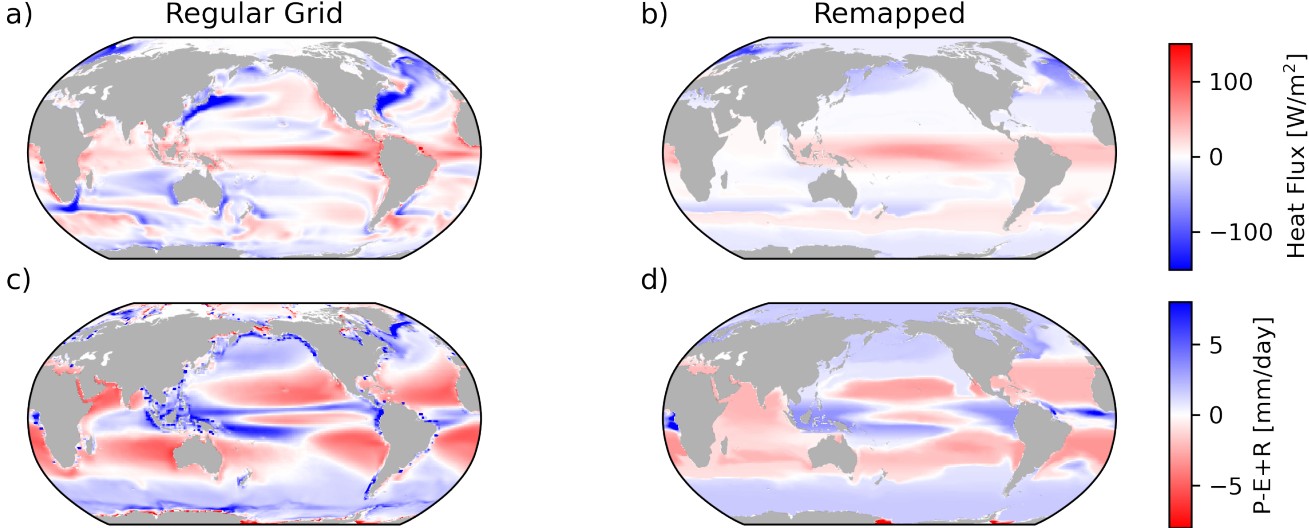

**Figure 5.** Time-averaged surface fluxes between the 'early' and 'late' periods in ACCESS-CM2, in the original model grid ($\mathbf{q}(x,y,t)$; (a) and (c)) and remapped onto the $2^n \times 9$ water masses as defined by BSP in T-S space ($\mathbf{Q}_i^{prior}$; (b) and (d)). Note that the surface outcrop location of these watermasses, averaged over the entire 'early' period, is used for the remapping.

The known surface fluxes, $\mathbf{Q}_i^{prior}$, are mapped onto the finite water masses obtained from the BSP coarsening (see figure 5). As the outcrop area of the water masses is much larger than the original model grid, the resulting remapped surface fluxes are smoother than the raw fields, as shown in figure 5.

In the remainder of this section we will discuss three applications of the inverse method with the same tracer data but different priors for the tracer sources and sinks – Case 1: the true tracer sources and sinks from the numerical model; Case 2: the true numerical model sources and sinks with a bias added globally; and Case 3: prior sources and sinks set to zero globally.

### 4.1 Case 1: 'True' source and sink priors

When the true model fluxes are used for $\mathbf{Q}^{prior}$ ($\mathbf{bias} = \mathbf{0}$), the inverse method is able to find a solution for $g_{ij}$ which matches these priors with little $\mathbf{Q}^{adjust}$ necessary (Fig 6). Quantitatively, the standard deviation of the true fluxes ($STD(\mathbf{q}_{prior})$; the signal) is [17.6 W m$^{-2}$,1.57 mm/day] while the standard deviations of the adjustment ($STD(\mathbf{q}_{adjust})$ the error) is [9.6 $\times 10^{-3}$ W m$^{-2}$, $7.4 \times 10^{-5}$ mm/day], yielding a signal to error ratio of, at minimum, order 2000.

From the inferred transport matrix $g_{ij}$, the region-to-region heat and freshwater transport is determined using

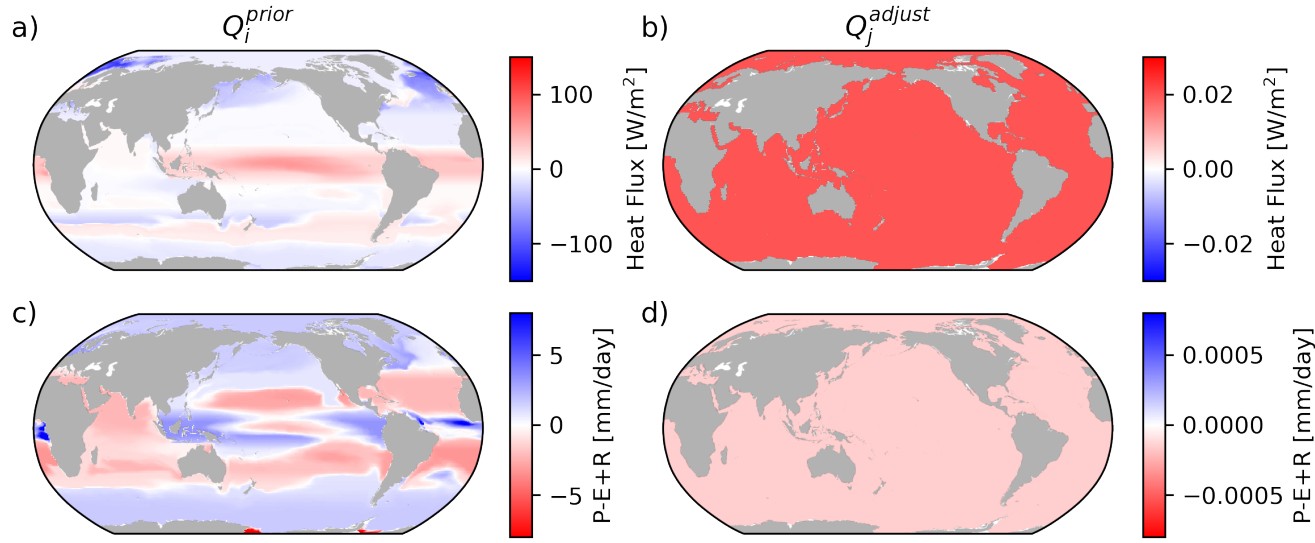

**Figure 6.** Case 1: Time-averaged surface fluxes between the 'early' and 'late' periods in ACCESS-CM2, remapped onto the $16 \times 9$ water masses as defined by BSP in T-S space ($\mathbf{Q}_i^{prior}$; (a) and (c)), and the inferred surface flux adjustment based on changes to the underlying ocean $T - S$ distribution ($\mathbf{Q}_j^{adjust}$; (b) and (d)). Note that the surface outcrop location of the water masses, averaged over the entire 'early' period, is used for the remapping. Note the difference in colourbar ranges between left and right panels

$$[\text{Heat transport}] = C_p \rho_0 \sum_{i=1}^{N} m_{0,i}(T_{0,i} + \mathbf{Q}_i^{prior}) g_{ij} \delta_{ij}; \tag{19}$$

$$[\text{Fresh water transport}] = -\rho_0/S_0 \sum_{i=1}^{N} m_{0,i}(S_{0,i} + \mathbf{Q}_i^{prior}) g_{ij} \delta_{ij}. \tag{20}$$

where $C_p$ is the heat capacity of sea water (3992.1 Jkg$^{-1}$K$^{-1}$), $\rho_0$ is a reference density (1035 kgm$^{-3}$) and $S_0$ is a reference salinity (35 g/kg). Above, $\delta_{ij} = 1$ if flow from $i$ to $j$ implied 'positive' transport across a region-to-region boundary (e.g. Northward across a zonal section) and $\delta_{ij} = -1$ if flow from $i$ to $j$ implies 'negative' transport (e.g. southward) and $\delta_{ij} = 0$ if water masses $i$ and $j$ are not in adjacent regions. We only consider region-to-region boundaries where the total mass transport is zero.

We compare the heat transport in ACCESS-CM2, inferred directly from model output, to our inverse estimate (based on 19) and the two match to within a standard deviation across the region-to-region boundaries of 17 TW in the Indo-Pacific and 16 TW in the Atlantic. Comparing the explicitly calculated fresh water transport in ACCESS-CM2 to our inverse estimate, we find that the two match to within a standard deviation of 0.14 Sv in the Indo-Pacific, and 0.014 Sv in the Atlantic (Figure 7).

It is reassuring that, when applied to consistent tracer source and tracer change data, an accurate solution is confirmed. We

now consider what happens when the prior source estimates contain biases.

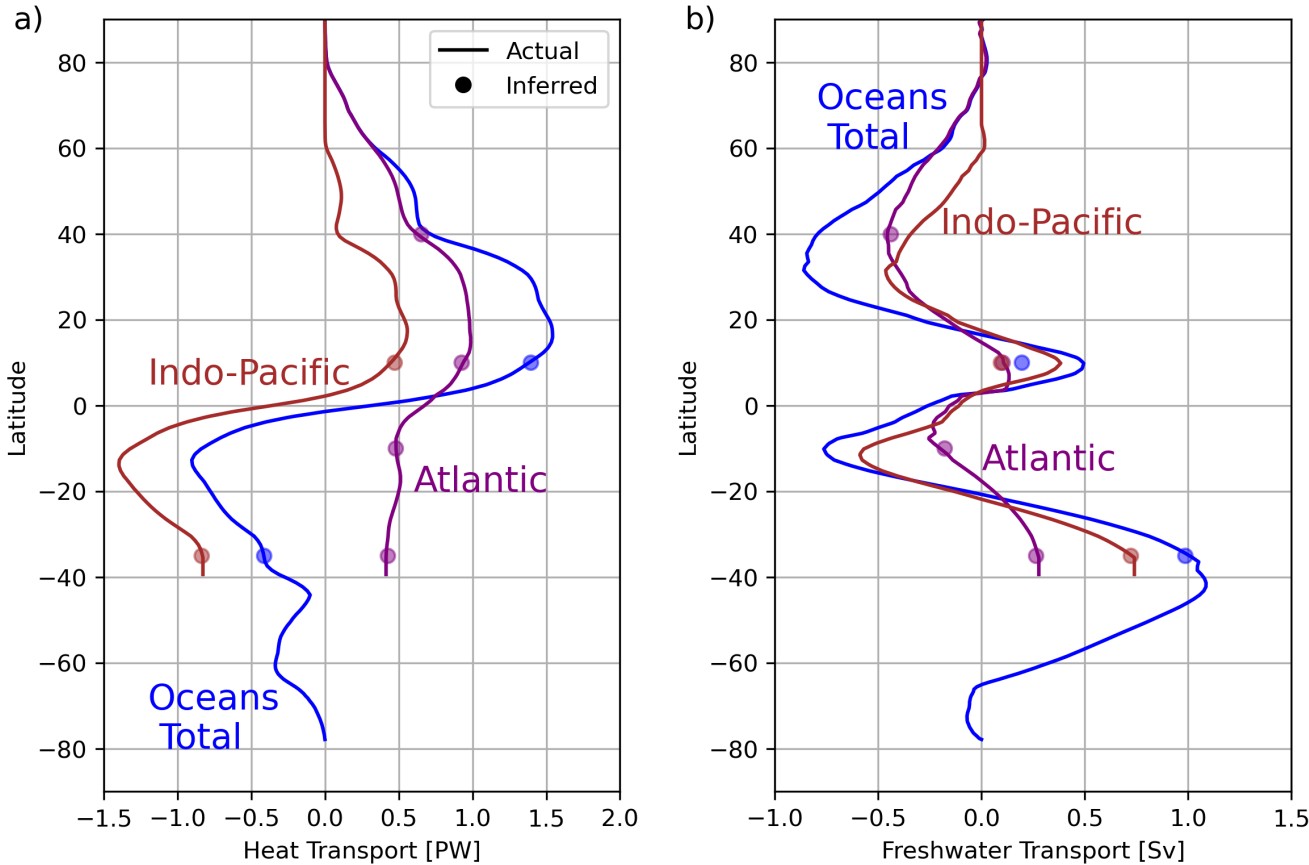

**Figure 7.** Case 1: Meridional a) heat transport and b) freshwater transport inferred from the transport matrix, $g_{ij}$ (dots, located at the boundary between adjacent regions), and from the surface fluxes and ocean heat/freshwater content change in the ACCESS-CM2 model (lines). We have omitted the same figure for Case 2 since the solution is indistinguishable.

### 4.2 Case 2: Spatially uniform biased source and sink priors

We add a constant offset to the air-sea fluxes of 5 W/m$^2$ for heat and 5 mm/day for fresh water over the entire data set (Fig.8). We then use the biased air-sea fluxes to determine $\mathbf{Q}^{prior}$ and feed this into our inverse model. The inverse model finds a solution for $g_{ij}$ and a $\mathbf{Q}^{adjust}$, via (17), opposing the bias to within a standard deviation of $2.9 \times 10^{-3}$ mm/day and $5.1 \times 10^{-2}$ W/m$^2$. The implied region-to-region heat transports of the inverse model with biased sources and sinks are virtually indistinguishable from the case without a bias, with a standard deviation that is within $1 \times 10^{-2}$ of the values reported for Case 1 (Fig.7).

This suggests the inverse model could be a useful tool to find a consistent, and potentially more realistic solution, in the presence of biased estimates of air-sea fluxes.

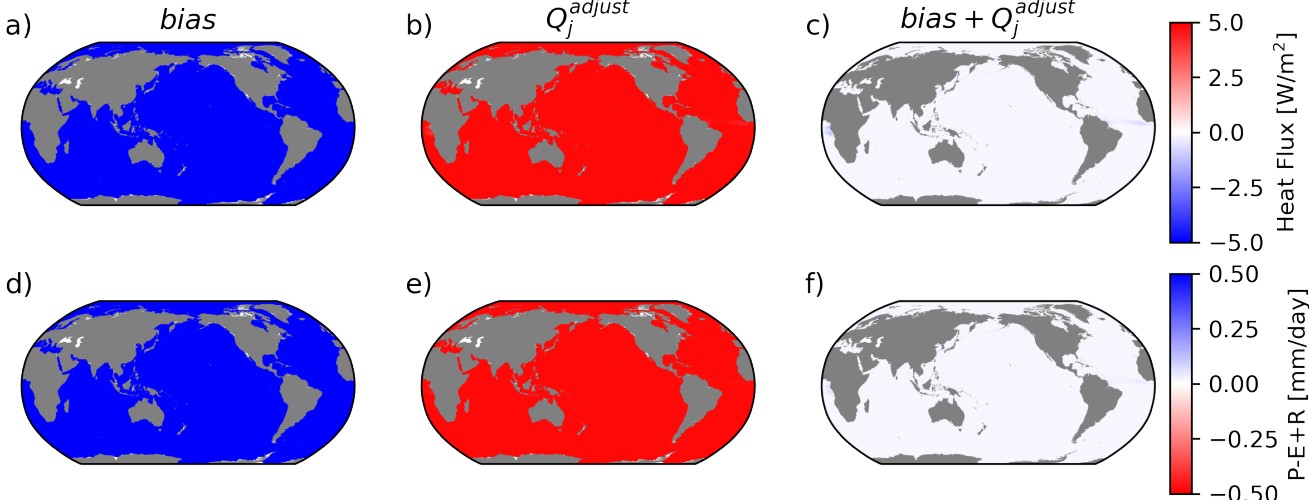

**Figure 8.** Case 2: Constant offset added uniformly to ocean surface fluxes ($bias$; (a) and (d)), the inferred adjustment based on changes to the underlying ocean $T - S$ distribution ($\mathbf{Q}_j^{adjust}$; (b) and (e)), and the sum of the two ((c) and (f)). Note that the surface outcrop location of the water masses, averaged over the entire 'early' period, is used for the remapping.

We now consider how the inverse method adjusts the prior fluxes when a reanalysis flux field (ERA5) is imposed as a prior, instead of the native model fluxes.

### 4.3 Case 3: Air-sea reanalysis-based source and sink priors

Case 3 offers a more practical test than cases 1 and 2 in that an incorrect, yet plausible, air sea flux field is used as the prior. We use observational estimates of heat and fresh water fluxes (from the ERA5 reanalysis; see Section 3) to determine $\mathbf{Q}^{prior}$. We expect differences between these fluxes and the known model fluxes since the model is not a perfect representation of the real climate. This mimics a scenario where we have good knowledge of $T$ and $S$ changes but air see fluxes have large biases which are heterogeneous. In this case, given we know the 'true' fluxes, we can see how well the inverse model does.

Figure 9 shows that ERA5 suggests warming across all ocean basins except the arctic relative to ACCESS, and a heterogeneous pattern of fresh water flux change, particularly at mid to low latitudes where rainfall is more intense on average.

The adjustment OTM finds in Case 3 is far more spatially homogeneous than the actual bias (Fig. 9, (b) and (e)). This is likely due to our cost function weights, which penalise adjustments in a per unit area sense. That being said, some basin scale pattern is captured well with less (more) heat (fresh water) adjustment in the arctic for example leading to accurate estimates of meridional transport between basins (Fig. 10).

Finally, we consider what the inverse method yields when we ask it to estimate the sources and sinks with priors set to zero.

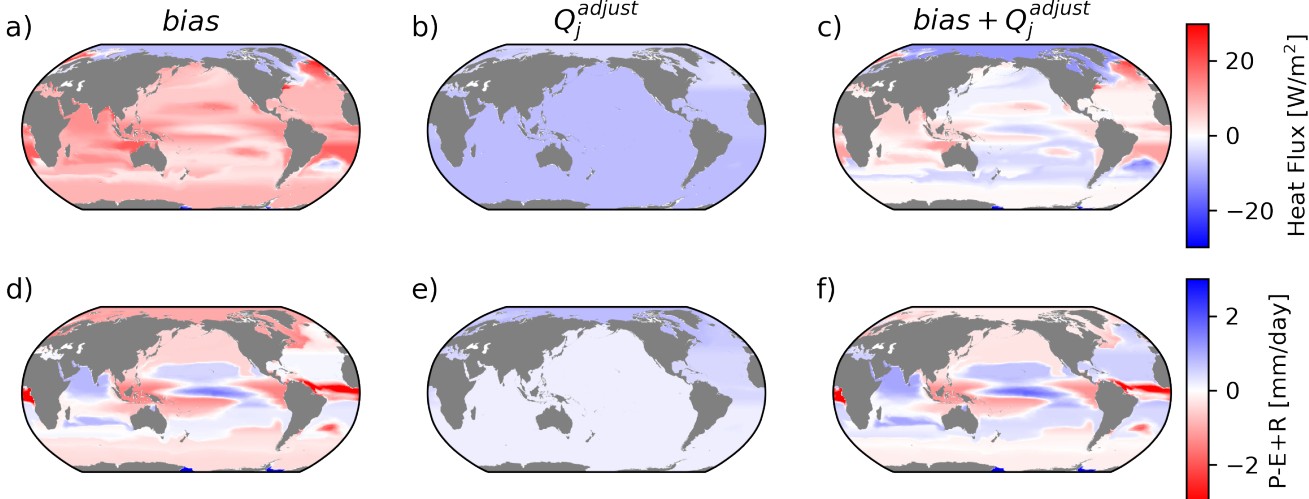

**Figure 9.** Case 3: The bias ((a) and (d)) is the difference between air sea heat (a - c) and fresh water (d - f) fluxes between ERA5 and ACCESS, (b) and (e) shows the inferred adjustment using OTM, and (c) and (f) shows the sum of the two. With the cost function in OTM minimising adjustments in a per unit area sense, it does not accurately correct the heterogeneous pattern of bias and favours a more spatially uniform adjustment.

## 4.4 Case 4: Zero source and sink priors

Cases 1 and 2 mirror toy examples 3 and 4 from Section 2, respectively. There, $\mathbf{Q}^{prior}$ effectively moved the water masses from their initial state to some intermediate state in tracer coordinates and then $g_{ij}$ moved them as close as possible to their final state, with $\mathbf{Q}^{adjust}$ providing the final adjustment. In our final case, we see how the inverse model responds to zero source/sink information, as in toy examples 1 and 2.

We run the inverse model, as in cases 1 and 2, but for $\mathbf{Q}^{prior} = 0$. The $\mathbf{Q}^{adjust}$ patterns represent the smallest necessary heat

and fresh water fluxes that can explain the model's water mass changes in conjunction with redistribution and mixing achieved by $g_{ij}$. Since the model is describing historical climate change, increases in ocean heat content and any increase in the variance of ocean salinity can not be described by $g_{ij}$ and are captured in $\mathbf{Q}^{adjust}$.

The resulting pattern of adjustments to the heat flux are approximately uniform across all oceans, except for polar regions (Fig. 11). In the inverse model solution, basin-scale anomalous warming/cooling patterns can be explained by redistribution via

$g_{ij}$. Only a small, near-uniform warming is required to complete the picture. The patterns of adjustment fresh water flux show net precipitation into relatively fresh regions of the globe such as the tropical pacific and sub-polar oceans and net evaporation over relatively saline regions such as the sub-tropical oceans and the majority of the Atlantic Basin. This is likely because greenhouse forcing in ACCESS is consistent with the 'wet gets wetter, dry gets drier' paradigm (Durack et al., 2012; Skliris et al., 2016) and the consequent changes in salinity can not be affected by mixing, which can only make fresh water salty and

salty water fresh (Zika et al., 2015b).

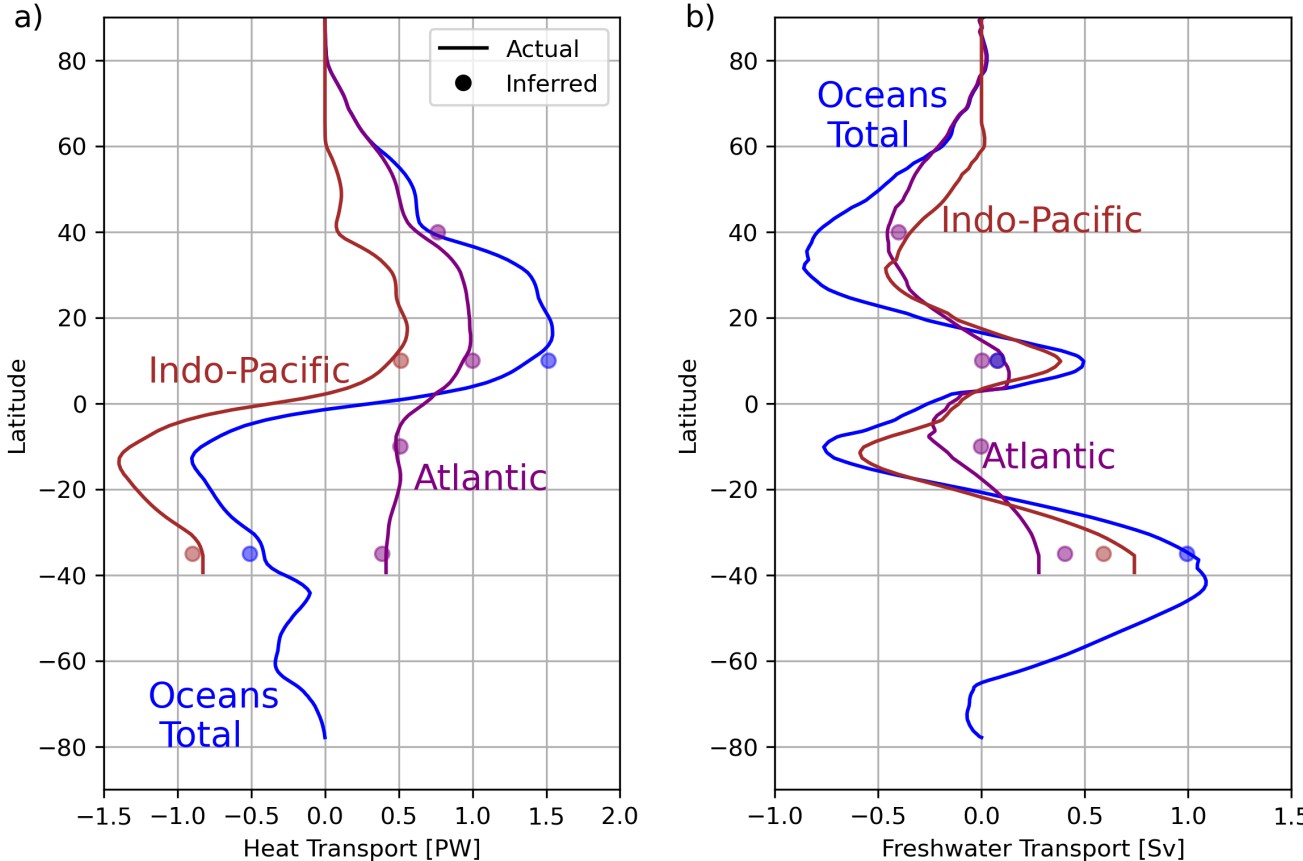

**Figure 10.** As in 8 but for Case 3.

The true air-sea fluxes warm the low latitudes and cool the high latitudes far more and this is balanced largely by heat transport and mixing represented by $g_{ij}$. Practically, a solution can always be added in which sources and sinks are balanced by the transport matrix while still satisfying our hard constraints (a 'homogeneous solution' in the language of differential equations) but in the case where $\mathbf{Q}^{prior} = 0$, such additions are penalised since the inverse method searches for the solution with the smallest root mean squared $\mathbf{Q}^{adjust}$. These results suggest that, without adequate priors, the inverse method cannot by itself accurately determine the correct total tracer sources and sinks.

Figure 12 summarises the results of the four cases at the basin scale. It shows the net $\mathbf{Q}^{prior}$ (if any), $\mathbf{Q}^{adjust}$, divergence of tracer transport described by $g_{ij}$, and the change in amount of tracer with time in each region.

Case 1 describes the true budget for the time period considered with the change with time and a small residual of the larger source/sink and divergence terms. Case 2 shows how a small adjustment to the sources and sinks compensates for an imposed error.

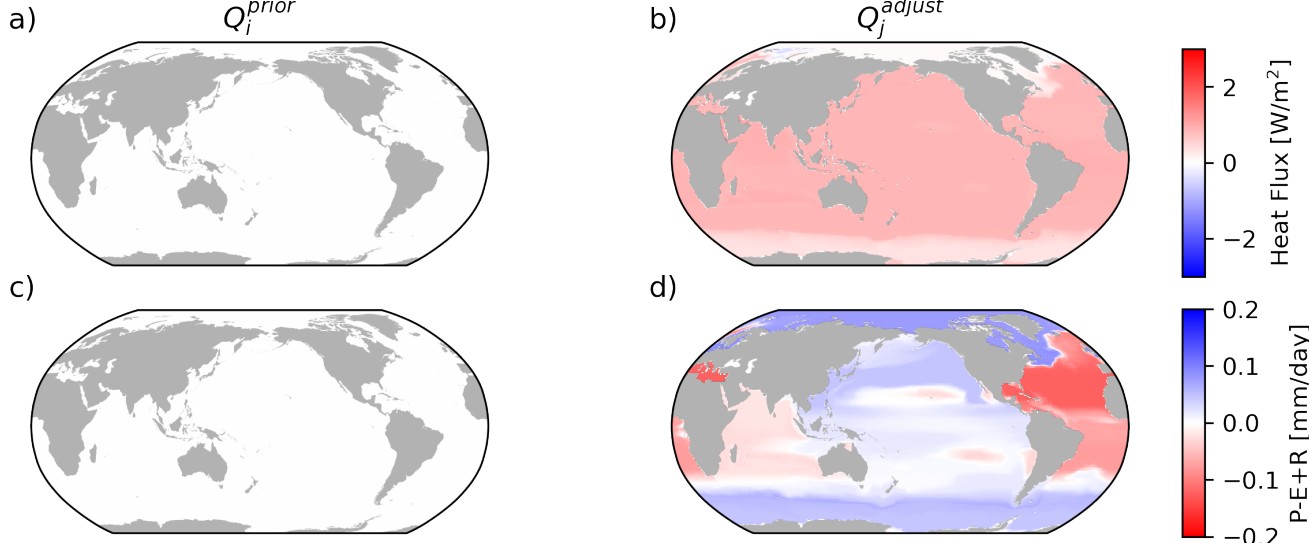

**Figure 11.** Case 4: Surface flux adjustment given no prior source/sink information ($\mathbf{Q}_i^{prior} = 0$; (a) and (c)), and the inferred surface flux adjustment based on changes to the underlying ocean $T - S$ distribution ($\mathbf{Q}_j^{adjust}$; (b) and (d)). Note that the surface outcrop location of the water masses, averaged over the entire 'early' period, is used for the remapping.

In Case 3 a spatially heterogeneous bias pattern is imposed. Despite the fact that the adjustment does not capture many of the finer geographical details, OTM does offer skill in balancing biases at the basin scale. Fig. 13 shows a comparison of the magnitudes of the added bias and adjustment in response for Case 3.

In Case 4, the implied net $\mathbf{Q}$ and tracer transport divergence are an order of magnitude smaller than in Cases 1 at the basin scale, since they are only required to describe the change rather than the large mean balances of sources/sinks and transport/mixing.

## 5   Discussion

Our assessment of the Optimal Transformation Method state estimation framework has not been exhaustive. Our aim has been 450   to describe the framework generally. In any future implementation, a number of choices can be made by the user, including:

1. The way water masses are defined both in space and time;

2. The way constraints are placed on the transport matrix $g_{ij}$ and priors are introduced; and

3. How adjustments of tracer sources/sinks and other variables impact the cost function.

For choice 1, we used binary space partitioning to objectively divide tracer space into discrete water masses. However, we 455   used conventional definitions of ocean basins to distinguish the water masses. OTM is not tied to either choice and alternative

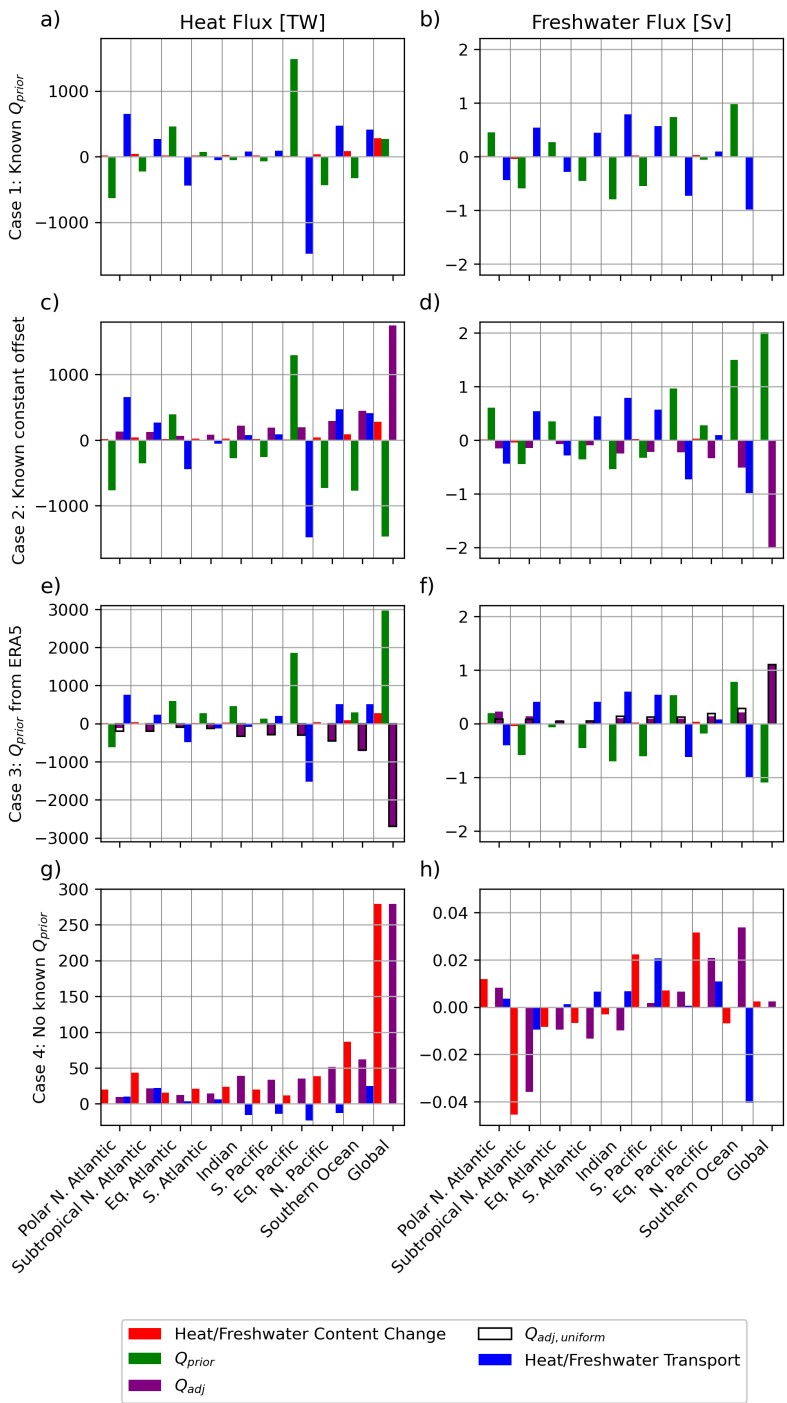

**Figure 12.** Terms in the heat and freshwater budgets for the four cases explored in this study. In this framework, Heat/Freshwater Content Change $= \mathbf{Q}_{adj} + \mathbf{Q}_{prior} +$ Heat/Freshwater Transport. For Case 1, the terms are indistinguishable from their 'true' values in the ACCESS model. For Case 3, unfilled bars show what the adjustment fluxes would be if they were globally spatially uniform

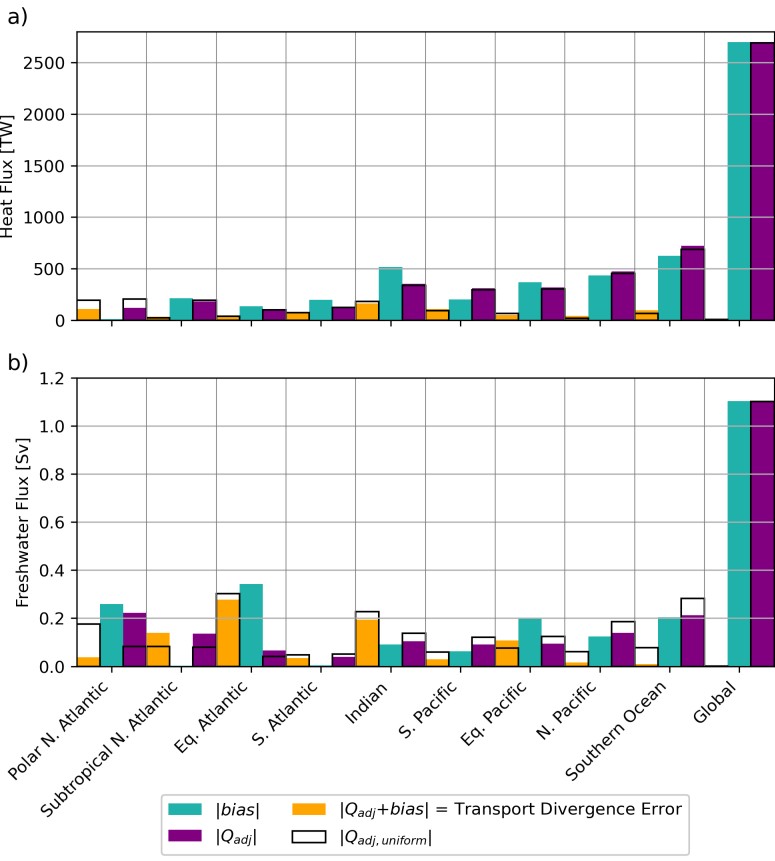

**Figure 13.** Comparison of bias and adjustment heat (a) and fresh water (b) fluxes for Case 3. In Case 3, ERA5 fluxes are used instead of the models true fluxes. Unfilled bars show what the adjustment fluxes would be if they were globally spatially uniform.

objective (e.g. machine learning based classification) and/or user-driven approaches (e.g. traditional water mass definitions) can be used. All that is required is that a set of water masses with tracer concentrations for two time periods (or a sequence of time periods) be defined and constraints be placed on their connectivity ($g_{ij}$).

For choice 2, we elected to give no prior information about the transport matrix ($g_{ij}$). Priors for this matrix or stricter constraints on it could be given based on numerical models or observations at key regional boundaries (such as the RAPID-MOCHA transect in the North Atlantic) and in key ocean gateways. Note, however, that $g_{ij}$ does not necessarily represent the conventional transport measured at a section. To illustrate this, consider a water mass in the subtropical North Atlantic with temperature $T_{0,i}$=20°C that is heated due to some air-sea flux with an implied warming over a 40 year period of $Q_i = 80$°C. Let us assume the state estimate tells us that 1% of this water mass travels northward into the sub-polar North Atlantic and mixes with 99% of the water contained in water mass $j$ (i.e. $g_{ij} = 0.01$ and $g_{jj} = 0.99$). Mathematically, the water can be viewed as

crossing the regional boundary at a temperature of $T_{0,i} + Q_i = 100°C$, as used in the calculation for the heat transport (19). A more plausible physical interpretation is that water from water mass $j$ is continually mixing with with water mass $i$. The state estimate does not describe where or when this mixing occurs, only that it occurred at some point between the early and late period. Hence, further work is required to determine how information about ocean overturning circulation can be used to constrain state estimates and likewise how the state estimate can inform us of the circulation.

For choice 3, in applications to observation based data, choices should be guided by the uncertainty in the underlying data. For example, we minimised the sources and sinks in a per unit area sense. OTM broadly did a better job than assuming a completely homogeneous pattern of change (see, for example, the comparison between unfilled and purple bars in Fig. 13) but did not accurately correct the biased pattern below the basin scale (Fig. 9). It could be that particular regions and/or components of the sources and sinks (e.g. precipitation) are more uncertain than others. These distinct uncertainties can be accounted for through the weight vector, $\mathbf{w}_j$.

An additional consideration that we have not explored here, is the choice of spatial and temporal resolution. We chose to compare between a mid-20th century and an early-21st century time period and force these with fluxes integrated between their mid point times. As the time period becomes longer, the fluxes can take the early water masses further and further apart in phase space (e.g. extending the vectors in Toy Example 2). These more extreme water masses are easier for OTM to mix together (via $g_{ij}$) to form late water masses. In the most extreme case, air sea fluxes would only be needed to translate global centre of mass of the distribution in tracer space, and $g_{ij}$ could account for all spatial variation in transformations. In that case, and in the absence of additional constraints on $g_{ij}$, OTM would have no skill in correcting spatial variations in tracer sources and sinks and would only correct the global mean.

## 6    Conclusions

We have presented a state estimation framework based on water mass theory, termed the Optimal Transformation Method. The framework enables the framing of inverse problems where ocean transport and tracer sources and sinks are optimally adjusted to define a self-consistent description of ocean change. We have used temperature and salinity data from a numerical climate model responding to historical natural and anthropogenic forcing over the past half century to test one application of the framework.

The Optimal Transformation Method draws on concepts in water mass transformation, water mass analysis and ocean tracer transport theory. What results is a set of equations describing how the ocean's multi-variate water mass distribution varies in time. These equations, combined with a transparent set of physically based constraints, allows for the definition of an inverse problem where a solution can be optimised based on deviations from priors.

We implemented an inverse method where the change in ocean state was known, ocean transport is unknown and deviations from prior estimates of tracer sources and sinks were minimised. When given 'true' heat and fresh water fluxes, the inverse solution found a state with near zero deviation from those priors. Likewise, when given fluxes with a constant bias added, the method reduced the error from 27.7% to 1.0% for heat flux and from 29.0% to 1.1% for fresh water flux. When given fluxes

with spatially heterogeneous errors, the method did a better job of correcting for that error at the basin scale than a constant compensating offset.

The methods presented may be a useful complement to existing state estimation approaches, having the advantage of being relatively simple (for example, when compared to numerical ocean models and ocean data assimilation platforms) and computationally cost efficient. In particular, the Optimal Transformation Method has shown promise for finding corrections to air-sea fluxes of heat and fresh water so that they plausibly describe the changing ocean state. This implies that the method, leveraged with observations, can help to refine observationally-based estimates of the net heat and fresh water flux imbalance in the climate system.

*Code and data availability.* The coarsened data from the historical ACCESS-CM2 simulation, as well as the scripts which carry out the optimisation, calculate flux budgets and plot surface flux maps are archived on Zenodo (Zika and Sohail, 2023). A working copy of the code and data is also available on GitHub (url: https://github.com/taimoorsohail/ACCESS_OTM.git).

*Author contributions.* The Optimal Transformation Method was conceived by JDZ and the concept was developed between JDZ and TS over a number of years. JDZ completed an initial proof of concept code which TS then developed into a an efficient working code base. TS carried out all the numerical calculations and prepared all the visualisations shown. Both JDZ and TS contributed to the text.

*Competing interests.* The authors declare no competing interests.

*Acknowledgements.* The authors would like to acknowledge Dr Neill Mackay who has tested some of our code and helped us improve this work. This work was supported by the Australian Research Council through Grants DP190101173 and SR200100008.

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
