# Peer review of "An optimal transformation method for inferring ocean tracer sources and sinks"

_EGUsphere, 2023_

## Referee Comment (RC1)

**Review of: An optimal transformation method for inferring ocean tracer sources and sinks**

Authors: J. Zika and T. Sohail, 2023, manuscript for EGUsphere

This paper presents a new framework to estimate oceanic tracer transport and refine measurements of air-sea flux. The framework combines ideas from water mass transformation theory and tracer transport models. The authors show several idealized, and then three more practical, applications of the framework. Currently, measurements of air-sea flux and tracer transport are uncertain, and the technique presented is a clever and promising new tool for addressing this complex and important challenge in climate science. I find the paper is well written and generally very clear for such a technical topic. I also think it's a good fit for the journal, and I expect the framework will be of interest to the community, given its broader implications for reducing uncertainty in estimates of air-sea fluxes and energy imbalance in the climate system

However, before publication, I would suggest some minor revisions. My main suggestions are that, while the paper is generally well written and clear, there are opportunities to make the method and results more physically intuitive in geographical space. Additionally, as someone who hasn't myself worked on inversion/optimization problems very deeply, I wasn't clear on the motivation behind some assumptions in the paper. Specifically, it would be useful for the authors clarify which assumptions are physically or dynamically motivated, versus those made because that's just how these problems are usually set up (i.e., because some assumptions need to be made to solve for a non-unique solution). I will give specific cases of both of these points, and more minor edits and questions, below.

Specific points:

L134: I am confused about the point regarding the EMD. The EMD method isn't being used here, right (there is no "$d$" in the equation)? I think it's the wording; what is "approach" referring to specifically? Is $Q_{flux}$ the equivalent of the minimized cost, or of d, or of g in (3)?

L137: I'm a bit unclear on why $g_{ij}$ acts on $Q_{ij}$. Is it because Q is the flux into watermass $i$, on its transit to $j$, but only the $g_{ij}$ frcation of $i$ makes it into $j$? It would help to add a sentence clarifying this here, because the previous section implies that Q is the total flux induced tracer change from $i$ in $j$ (i.e., as if the fraction of $i$ in $j$ was already accounted for).

L148: I'm confused about the motivation for this set up. If it isn't well known, why can we hope $Q_{adjust}$ to be small? Will this force the major changes in tracer between water masses to effectively be put into $g_{ij}$? More broadly, is this based on a physical or practical reasoning, or is it arbitrary that you minimize $Q_{adjust}$ and not $g_{ij}$? Do we assume that, in practice, the surface fluxes will be easier to guess at than the mixing? It would also help to lay out the basic idea here, or in the section on the EMD before (i.e., explain that two things are not well known, or known at all, and an established approach is to minimize the deviation in one from its prior, because the solution to (6) is non-unique). Also "non-mixing cost" isn't super clear. I would suggest explaining that or renaming.

L154) Here reiterate what solving for $g_{ij}$ means physically... solving for the minimised mixing and advection?

Eq. 8) Here I would suggest writing instead: $\sum g_{ij}Q_{adjust} = C_{1,j} ... - \sum g_{ij}Q_{prior}$ , and then could say in the text that the total flux experienced in transit is : $\sum g_{ij}(Q_{adjust} + Q_{prior})$ or something. I suggest this because in the upcoming sections, you really only talk about $Q_{adjust}$ adjust (not $Q$), so it is nice to have an equation to refer back to specifically. Also, to be as clear as possible for the following sections, I would suggest clarifying that the method is to use (7) to calculate g and then (8) to solve for $Q_{adjust}$.

L167) Related to my point above, it's not physically obvious to me why it might make sense to adjust the fluxes minimally in a per unit area sense. Could you explain? Also, it would be good to note here that you do use this assumption in some of the following examples (Eq. 15, etc).

L204) Clarify that "no cost" means the solution can be achieved with mixing alone, i.e. no adjustment to the fluxes; similar at L211.

L244) Are the dates here backwards?

Fig 3) Maybe draw the bin boundaries from Fig. 4 on these distributions (i.e., the boxes)?

L270) ...of equal volume "globally" (add globally to make subsequent statement about being different masses in each basin clear). Also, would be good to say here this is a Boussinesq model, since you are using mass and volume interchangeably.

L272) Sentence "we partition" the 16 water masses: here would be good to clarify that it becomes 16 water masses because different water masses with the same T-S properties exist in each basin.

Fig. 4: in each of the "14" bins. Also, I'm confused by the points. How are you calculating them?

Eq. 9) I had a lot of trouble with the concept of $A_i$ here and its use in the following equations. Why is the time integral over the midpoints of the early and late periods, not the endpoints of the early period, if it's the outcrop area of watermass $i$? Instead, $A_i$ is the average area of the outcrop of the initial watermass the time it transits to the final watermass (right)? I think that needs to be explained better since it's not immediately intuitive to me why you use this as the outcrop area over which $Q$ acts. Also, what is the zero here in $\Omega(x, y, 0, t)$? A basin tag?

L280-281) Perhaps recall here that the following hard constraints are extensions of earlier equations (Eq. 2, etc), representing mass conservation, total tracer conservation, and transport speed/likelihood constraint.

L291) Isn't this using Eq. 15, not 7?

L295) Could you explain this a bit more (i.e., "instead of the average area over time, we skew it towards the smallest possible positive value?")

L300/Eq. 17) I'm still a bit hung up on $\Omega$ here. I think a schematic of one parcel $i$'s physical journey to $j$, in geographical space, would be helpful. This could include a diagram of the area we are using as $\Omega$ and $A_i$. I think in general, this schematic would be helpful earlier in the paper to gain physical intuition of the method.

L312) Is there a reason that $Q_{adjust}$ is constant (is it hard to get out spatial patterns)?

L335) How is up or downstream calculated? Using the streamfunction/velocity? Maybe this was explained and I missed it?

L350) I feel that this is an important point and could be highlighted more. Essentially, this technique could help provide a better estimate of the net radiative imbalance in the climate system, which is hard to do!

Fig 7) Are the dots at the boundaries of regions? Where are they coming from? Is it possible to be more continuous? Also, this is case one and two, right?

L361) "polar regions" – refer to Fig. 9 here.

Fig 8) Mention which case is shown here.

L365) Do you understand why the freshwater is more successful as a non-uniform pattern? Does the fact that the heat fluxes are minimized as a uniform pattern mean that the optimization problem might be set up imperfectly? Not the you need to do redo it, but it would help if you could mention of why this is so, if you have intuition about it.

L370) I'm not sure what "increases the cost function" means. Do you mean that it "is not a minimum of the cost function?"

L390-400) I'm curious about other dynamical ways to constrain the problem. Would it be possible to include a feature of the weights that discounts net volume transport across the strong meridional buoyancy gradients (for instance, incorporating the tendency for advection to be along-isopycnal)? This doesn't need to be discussed in the paper, I'm just curious. In the text, however, I again suggest expanding on why the assumptions regarding the prior knowledge of g and Q, were made here from a dynamical standpoint.

Fig 10) Could a panel be added with the truth? This is not required if it's really complicated... (and truth would only include the total fluxes, content change, and transport, I think). But would be nice to compare to.

L417) I would again suggest highlighting the point here that "this implies that the method, leveraged with observations, might help to refine observationally-based estimate of the net heat flux imbalance in the climate system." Or something....

Grammatical edits:
L31) Comma before "which"
L54) in space "and time"
L88) add quotations around "conservative"
L131) comma before "which"
L175) comma after "implausible"
L264) comma before "which"
L265) I would suggest replacing "volume" with "mass" since you used mass before..
L360) Remove second "of"
L361) "Adjusted" (?)

---

## Referee Comment (RC2)

**Review of "An optimal transformation method for inferring ocean tracer sources and sinks" by Zika & Sohail for EGUsphere.**

The paper presents a new approach (Optimal Transformation Method), rooted in water mass transformation methods, to infer changes in tracer distributions in the ocean interior as a result of ocean transport (circulation and mixing) and tracer sources/sinks. The novelty of this method is that it allows to separate the effect of air-sea fluxes, which often have biases, and mixing; this separation is not usually allowed by other inverse techniques. Also, the OTM method is not based on a steady state ocean circulation assumption, hence allowing to investigate changes in the ocean circulation.

The authors present an application of this new framework to a historical numerical model, after discussing the framework details with idealised case scenarios. This new framework is an interesting new approach, complimentary to other existing methods. The paper is overall very well written and some of the technical aspects of the methodology are clearly explained. I think this manuscript fits well in EGUsphere. Before publication, I think there are some aspects of the paper that need clarification. These are overall minor revisions, discussed below.

Comments:

- Line 48: More than in GF, it seems to me the method is rooted in transport matrix and water mass theory..?

- Line 118 (and following discussion at lines 123-127): Perhaps it might be worth to introduce a definition of a water mass? In the usual definition, which might not apply here, a water mass is defined as a "body of water with common formation history", or a "body of water whose conservative properties are set by a single, identifiable process (and altered only by mixing)". The conservative properties defining a water mass are most often set at the surface (some non-conservative properties can be acquired in the interior, e.g. an oxygen minimum, but most often that is not the case). Hence, why we usually describe properties in the interior as a linear combination of surface properties. My understanding is that in the OTM approach, the definition of a "water mass" is looser than the convention (e.g. line 118: using the definitions above, the mix of two known water masses is not a new, separate water mass), so it might be worth stating this difference from a conventional definition.

- Line 134: The reference to EMD is a bit confusing. Maybe I got it wrong, but my understanding is that $Q_{i,j}$ is the distance in tracer space between the early and late water masses due to sources/sinks. If that's the case, it might be beneficial

to write that explicitly in the definition of Qi,j at line 134, so that the following statement might become less confusing. Or rephrase/expand on the EMD reference (also because you are not using the EMD in the OTM, right?)

- Line 137: I think clarifying the point above about Qi,j definition would help to better understanding Eq.5. I was initially confused about gi,j acted on Qi,j.

- Line 148: What is the reasoning here? The previous statement says that the confidence in Qi,j is low, hence the confidence in the prior is low, correct? Why should the solution assume that Qadjust is small?

- Line 150: I might have missed it, but why is the cost function in eq. 7 called called "non-mixing cost"? Also, it was not until I read the Results section that it became clear that the steps are to (i) solve for gi,j in (7) and (ii) then calculate Qadjust in (8). I would suggest to state more clearly here.

- Fig 4: I am a bit confused by this figure. If I understand correctly, first the ocean is split in 16 T-S groups of equal global volume, and fig. 4 shows the volume of each of this groups in the 9 geographical regions considered. So, if we were to sum up the volumes across all nine regions per each water mass, we should retrieve the same volume? I might be reading this wrong, but I do not see this in Fig.4. Take for example the water mass defined by T[-2:5] and S[30:34.5] approx (bottom left box). This water mass has a relatively low volume compared to other water masses in almost all regions (but N. Pac, perhaps). It overall seems to be orders of magnitude lower than the volume of the water mass defined by T[-2:4] and S[34.5:35.5] approx. Again, I might be reading this wrong, so some clarification would be appreciated.
Also: (1): it would be useful to add these boxes in Fig.3 and use the same x and y intervals and spacing, if possible; (2) only 14 of the water masses are visible in Fig.4, maybe change the axis to improve visualisation?

- Eq. 9: Ai is not the outcrop area at the early stage (right?), which is would one would most likely assume. Is it the outcrop area while transitioning between early and late periods? Some clarification would be useful. Also, should \Omega_i be \Omega_i(x,y,t) only (also in Eq. 19)?

- Line 311-312: Why don't you attribute different adjustments, but Qadjust is the same for all i?

- Fig. 6: Perhaps change the colorbar for Qadjust, so that they are not just blank? Or remove the figure and just use the signal to noise reported (lines 328-329) to make the point that Qadjust << Qprior?

- Fig. 7: The number of points where transports can be inferred is limited by the number of regions selected, correct? Also, perhaps add the inferred transports

for Case 2 and show that they are indistinguishable and change caption to mention both Case 1 and 2?

- Line 344: Can you add a justification of why you selected 5 W/m2 for the heat flux bias, and 5 mm/day for the fresh water flux bias? Why not larger/smaller (well, I guess larger would be more interesting) biases? And why not a percentage of the signal, rather than a fixed amount? And what if the biases were not uniformly positive/negative? Maybe I missed the point, but how could a fixed Qadjust reflect a mix positive/negative biases?

- Line 361-363: I think I am off here, comparing apple and oranges, but how does the result for the heat flux compare with the redistributed vs added heat? Can we interpret fig.9 (for the heat flux changes) as an indication that most of the ocean heat content changes are described by redistributed heat (gi,j explaining most of it), and only part of the changes are caused by added heat?

Minor comments:
- Line 18: Estimates (capital E)
- Line 19: Remove "However"?
- Line 38: Delete "[" at the end of the line
- Line 70: "properties" misspelled
- Line 151: Add reference to section: "where wj is a relevant weighting (see section 2.5)".
- Line 173: "early" and "late" in wrong order.
- Fig 3: colorbar label has kg spelled differently in the same label (Kg and kg)
- Line 291: Eq 15 (and not 7)?
- Line 339: we "find" (verb missing)
- Line 360: two "of"
- Line 361: add reference to fig. 9. Also, maybe change the colorbar for the adjusted heat flux?

---

## Author Comment (AC1)

Response to reviewer suggestions for egusphere-2023-1220 - "An optimal transformation method for inferring ocean tracer sources and sinks" by Jan Zika and Taimoor Sohail.

Responses are in blue with proposed edits indented.

Reviewer 1

This paper presents a new framework to estimate oceanic tracer transport and refine measurements of air-sea flux. The framework combines ideas from water mass transformation theory and tracer transport models. The authors show several idealized, and then three more practical, applications of the framework. Currently, measurements of air-sea flux and tracer transport are uncertain, and the technique presented is a clever and promising new tool for addressing this complex and important challenge in climate science. I find the paper is well written and generally very clear for such a technical topic. I also think it's a good fit for the journal, and I expect the framework will be of interest to the community, given its broader implications for reducing uncertainty in estimates of air-sea fluxes and energy imbalance in the climate system.

However, before publication, I would suggest some minor revisions. My main suggestions are that, while the paper is generally well written and clear, there are opportunities to make the method and results more physically intuitive in geographical space. Additionally, as someone who hasn't myself worked on inversion/optimization problems very deeply, I wasn't clear on the motivation behind some assumptions in the paper. Specifically, it would be useful for the authors clarify which assumptions are physically or dynamically motivated, versus those made because that's just how these problems are usually set up (i.e., because some assumptions need to be made to solve for a non-unique solution). I will give specific cases of both of these points, and more minor edits and questions, below.

We sincerely thank the reviewer for these encouraging comments. Our responses are below in blue.

Specific points:

L134: I am confused about the point regarding the EMD. The EMD method isn't being used here, right (there is no "d" in the equation)? I think it's the wording; what is "approach" referring to specifically? Is $Q_{\text{flux}}$ the equivalent of the minimized cost, or of d, or of g in (3)?

We recognise that mentioning the work using EMD is such detail was not necessary or useful and have removed most of that content. The paragraphs now read:

> In Zika et al. (2021), we solved for $g_{ij}$. by minimizing the amount of warming and cooling water masses had to undergo, in a root mean square sense, to achieve the observed change in water mass distribution in temperature and salinity coordinates (see also Evans et al. (2014)). Using that approach we are not able to make use of observational estimates of air sea heat and fresh water fluxes nor were we able to impose physics based constraints on mixing driven transformations.
>
> Here we present a method where the influence of sources and sinks of tracer, circulation and mixing are considered separately, which we call the Optimal Transformation Method (OTM). We now discuss how mixing and tracer sources and sinks can drive transformation and modify the water mass distribution in tracer space.

L137: I'm a bit unclear on why $g_{ij}$ acts on $Q_{\mathrm{surf}}$. Is it because Q is the flux into watermass *i*, on its transit to *j*, but only the $g$ij fraction of *I* makes it into *j*? It would help to add a sentence clarifying this here, because the previous section implies that Q is the total flux induced tracer change from *i* in *j* (i.e., as if the fraction of *i* in *j* was already accounted for).

We recognise that this was not sufficiently clear. We have attempted to clarify the above point with the following text modifications:

The fraction of our $i$th early water mass which is transported to the $j$th late water mass can be subjected to a source or sink of tracer on its route from one to the other. We represent this source as an implied change in tracer concentrations $\mathbf{Q}_{ij}$ along the Lagrangian path taken by the fraction of water $g_{ij}$. That is, the fraction of water ($g_{ij}$) that leaves the early water mass $i$ with tracer concentration $\mathbf{C}_{0,j}$ can be thought of as having been changed to concentration $\mathbf{C}_{0,j} + \mathbf{Q}_{ij}$ by the time it arrives at late water mass $j$.

The late water mass $j$ is formed from the mixture of all the fractions of $g_{ij}$ modified along their respective paths such that its tracer concentration is

$$\mathbf{C}_{1,j} = \sum_{i=1}^{N} g_{ij} \left( \mathbf{C}_{0,i} + \mathbf{Q}_{ij} \right). \tag{4}$$

This provides a complete description of water mass change: the late water masses ($\mathbf{C}_{1,j}$) are formed as the linear combination of fractions ($g_{ij}$) of the early water masses ($\mathbf{C}_{0,i}$), each modified on route by sources and sinks ($\mathbf{Q}_{ij}$).

L148: I'm confused about the motivation for this setup. If it isn't well known, why can we hope $Q_{\mathrm{flux}}$ to be small?

Will this force the major changes in tracer between water masses to effectively be put into $g_{ij}$?

More broadly, is this based on a physical or practical reasoning, or is it arbitrary that you minimize $Q_{\mathrm{flux}}$ and not $g_{ij}$? Do we assume that, in practice, the surface fluxes will be easier to guess at than the mixing? It would also help to lay out the basic idea here, or in the section on the EMD before (i.e., explain that two things are not well known, or known at all, and an established approach is to minimize the deviation in one from its prior, because the solution to (6) is non-unique).

Thank you for these suggestions. We have added the following text:

In practise, we do not know any of the 4 terms in (4) with certainty for any tracers in the ocean. We can, however, frame (4) as an inverse problem, and adjust the terms within it to find solutions under certain constraints. Many different strategies could be employed depending on the confidence of the user in the different terms and constraints. We will develop and implement one approach we consider relevant to understanding recent multi-decadal changes in ocean temperature and salinity.

For heat and salt, we consider there to be relatively good confidence in observational estimates of $\mathbf{C}_{1,j}$ and $\mathbf{C}_{0,i}$, poorer confidence in estimates of $\mathbf{Q}_{ij}$ and poor to no confidence in estimates of $g_{ij}$. The concentrations $\mathbf{C}_{1,j}$ and $\mathbf{C}_{0,i}$ can be derived from ocean temperature and salinity analyses (e.g. (Good et al., 2013)). These come with substantial uncertainties (Cheng et al., 2022; Stammer et al., 2021), but have the benefit of essentially being mappings of directly observed quantities. The source/sink term $\mathbf{Q}_{ij}$ can be inferred from air-sea flux products but these come with larger uncertainties. For example, heat content changes derived from temperature analyses vary by order $0.1 \mathrm{W/m^2}$ (e.g. $0.05 \mathrm{W/m^2}$ for 1958-2019, Cheng et al., 2022) while those derived from accumulated air-sea heat fluxes typically have biases of order $1 \mathrm{W/m^2}$ (e.g. $4 \mathrm{W/m^2}$ for 1993-2009, Valdivieso et al., 2017). Finally, we know of no direct way of deriving $g_{ij}$ from observations. Indeed, $g_{ij}$ could be derived from a data constrained numerical model, but that would imply it is indirectly derived from the same data used for $\mathbf{C}_{1,j}$, $\mathbf{C}_{0,i}$ and $\mathbf{Q}_{ij}$. We thus consider it reasonable to frame an inverse problem where $\mathbf{C}_{1,j}$ and $\mathbf{C}_{0,i}$ are considered 'known', priors for $\mathbf{Q}_{ij}$ are provided and $g_{ij}$ is merely constrained to obey laws of physics.

Also "non-mixing cost" isn't super clear. I would suggest explaining that or renaming.

We changed it simply to [Cost] since we have removed the EMD case from earlier.

L154)Here reiterate what solving for $g_{ij}$ means physically...solving for the minimised mixing and advection?

We have added:

Physically, solving for $g_{ij}$ using (6) implies we modify the early water masses with the prior source/sink estimates, then find the geographical rearrangement and mixing of those modified water masses that gets us as close as possible to the later water masses.

Eq. 8) Here I would suggest writing instead: $\sum g_{ij} Q_{adjust} = C_{1,j} - ... \sum g_{ij} Q_{prior}$

Thank you. Done.

and then could say in the text that the total flux experienced in transit is: $\sum g_{ij}(Q_{prior} + Q_{adjust})$ or something. I suggest this because in the upcoming sections, you really only talk about $Q'_{(ij)*}$ adjust (not $Q$), so it is nice to have an equation to refer back to specifically.

We have added:

The accumulated tracer source following the fluid motion from early water mass $i$ to late water mass $j$ is then $\mathbf{Q}_{ij}^{prior} + \mathbf{Q}_{ij}^{adjust}$.

Also, to be as clear as possible for the following sections, I would suggest clarifying that the method is to use (7) to calculate g and then (8) to solve for $Q_{adjust}$.

Indeed. We have added:

To recapitulate, we have described a method where we find the optimal transport matrix $g_{ij}$ using (6), and then, from this, we find the adjustment required to tracer sources and sinks using (7). We call this an Optimal Transformation Method (OTM) since we are looking for the optimal way in which the waters can be transformed to describe the evolving ocean state given our physical constraints.

L167) Related to my point above, it's not physically obvious to me why it might make sense to adjust the fluxes minimally in a per unit area sense. Could you explain? Also, it would be good to note here that you do use this assumption in some of the following examples (Eq. 15, etc).

Indeed. The paragraph now reads

The purpose of $\mathbf{w}_j$ is to favour solutions where the source and sink adjustments are more likely. One case where this is apparent is for tracers with little or no interior source or sink such as conservative temperature (essentially a tracer of heat), salinity (a tracer of fresh water) and anthropogenic tracers such as chlorofluorocarbons. For such tracers, it makes sense to not allow (or at least heavily penalise) fluxes into tracer sources in water masses that do not outcrop. More-over, if the flux of tracer per unit area at the sea surface had a known uncertainty, this could be used to derive the weights as the product of the uncertainty per unit area and the area. This way, adjustments to the tracer sources would incur a higher costs in (6) for water masses that have a small outcrop and/or have low uncertainties in the fluxes over that outcrop. In our toy examples and our application to data from a climate model, we will consider only the case where the uncertainty in the fluxes are the same in a per unit area sense so that the weights are proportional to the inverse of the area.

L204) Clarify that "no cost" means the solution can be achieved with mixing alone, i.e. no adjustment to the fluxes; similar at L211.

We have changed the paragraph to read:

prior sources/sinks such that $\mathbf{C}_{0,1} + \mathbf{Q}_{1j}^{prior}$=[0,34.6], $\mathbf{C}_{0,2} + \mathbf{Q}_{2j}^{prior}$=[4,35], $\mathbf{C}_{0,3} + \mathbf{Q}_{3j}^{prior}$=[0,35.4] for all $j$. A solution then exists with no cost, according to (6). That is, a valid solution can be found with mixing alone. This occurs when $g_{ij} = 0.5$ for $i = j$ and $g_{ij} = 0.25$ otherwise (as in the pure mixing case). In this solution, the sources and sinks expand the triangle, and according to the transport matrix, the water masses are then mixed together, contracting the triangle to achieve an unchanged water mass distribution.

L244) Are the dates here backwards?

The dates are correct.

Fig 3) Maybe draw the bin boundaries from Fig. 4 on these distributions (i.e., the boxes)?

This has been done.

L270) ...of equal volume "globally" (add globally to make subsequent statement about being different masses in each basin clear).

Added.

Also, would be good to say here this is a Boussinesq model, since you are using mass and volume interchangeably.

We have added:

> (since ACCESS-CM2's ocean component is Boussinesq, volume and mass are proportional to one another)

L272) Sentence "we partition" the 16 water masses: here would be good to clarify that it becomes 16 water masses because different water masses with the same T-S properties exist in each basin.

We have changed the text to simply say:

We recursively subdivide the $T - S$ distribution of the top 2000m of the global ocean in ACCESS-CM2 4 times to yield $2^4 = 16$ classifications of equal volume/mass globally (since ACCESS-CM2's ocean component is Boussinesq, volume and mass are proportional to one another). We further partition these 16 $T - S$ classifications into each of the 9 basins defined above over the full ocean depth. This produces what we define as our 144 'early' and 144 'late' water masses. Each water mass has different

We felt it was clearer to introduce the words 'water mass' only when we got to the T-S and geographical partition.

Fig. 4: in each of the "14" bins. Also, I'm confused by the points. How are you calculating them?

The figure caption now reads:

**Figure 4.** Volume (colours) and mean $T$ and $S$ of each of the 16 bins in each of the 9 basins analysed in the 'early' period. Each rectangle represents the range of $T$-$S$ values covered by one of the water masses. The colour of the rectangle represents the volume of water in that bin in that basin. Each white point contained within a rectangle is located at the average $T$-$S$ value of the water in that bin in that basin.

Eq.9) I had a lot of trouble with the concept of $A_i$ here and its use in the following equations. Why is the time integral over the midpoints of the early and late periods, not the endpoints of the early period, if it's the outcrop area of watermass *i*? Instead, $A_i$ is the average area of the outcrop of the initial water mass the time it transits to the final watermass (right)? I think that needs to be explained better since it's not immediately intuitive to me why you use this as the outcrop area over which $Q$ acts. Also, what is the zero here in $\Omega(x, y, 0, t)$? A basin tag?

For clarity, we have moved the definition of the average outcrop area to after the definition of the prior sources and sinks and added amended the text with the following:

*bias* is a bias we will introduce in some cases to see what effect incorrect air-sea flux data has on the inverse solution. The above time integral is from the midpoint of the early period to the midpoint of the late period since it is related to the change in average water mass properties between the two periods. (Integrating from the start of the early to the end of the later period would overestimate the sources and sinks.)

In our implementation of the optimisation (13) we aim to minimise the average adjustment to the tracer sources and sinks in a per unit area sense. For this reason we calculate the average outcrop area using the same integral limits as the sources and sinks such that

$$A_i = \frac{1}{t_1 - t_0} \int_{t_0}^{t_1} \iint \Omega_i(x, y, 0, t)\, dx\, dy\, dt. \tag{16}$$

L280-281) Perhaps recall here that the following hard constraints are extensions of earlier equations (Eq. 2, etc), representing mass conservation, total tracer conservation, and transport speed/likelihood constraint.

Thanks, we have added

The above enforce mass conservation (8-10), tracer conservation away from the surface boundary (11) and the inability of water to move further than the adjacent basin (12).

L291) Isn't this using Eq. 15, not 7?

Fixed

L295) Could you explain this a bit more (i.e., "instead of the average area over time, we skew it towards the smallest possible positive value?")

Indeed. We have changed the text to

ensures that changes to water masses that do not outcrop are achieved purely by redistribution and mixing. In one of the cases we will discuss below (where $\mathbf{Q}_i^{prior} = 0$), our optimiser does not find a feasible solution with this constraint when $A_i = 0$ for some $i$ values. In that case, we set a floor on those areas as the minimum non-zero $A_j$ found for all $j$. This was, in that specific case, the most permissive area constraint we could justify for the problem.

L300/Eq. 17) I'm still a bit hung up on $\Omega$ here. I think a schematic of one parcel *i*'s physical journey to *j*, in geographical space, would be helpful. This could include a diagram of the area we are using as $\Omega$ and $A_1$. I think in general, this schematic would be helpful earlier in the paper to gain physical intuition of the method.

When we first define \Omega and the outcrop area we have added:\

Each water mass has a corresponding 'mask', $\Omega_i(x,y,z,t)$ defining its geographical location with time ($\Omega_i = 1$ within the water mass and $\Omega_i = 0$ outside; $x$, $y$ and $z$ are latitude, longitude and depth respectively). The outcrop area of water mass $i$ at time $t$ is then, $\iint \Omega_i(x,y,0,t)dA$ and $A_i$ is the time average of that area (defined below).

It is not straight forward to draw a schematic of a moving set of water mass outcrops and we have chosen to leave this suggestion.

L312) Is there a reason that $Q_{\mathrm{adjust}}$ is constant (is it hard to get out spatial patterns)?

We have added

Because we only infer one adjustment flux per water mass, we are not able to infer more detailed variations in the flux over the spatial extent of the water mass outcrop.

L335) How is up or downstream calculated? Using the streamfunction/velocity? Maybe this was explained and I missed it?

Indeed, this was not clear. We now write:

salinity (35 g/kg). Above, $\delta_{ij} = 1$ if flow from $i$ to $j$ implied 'positive' transport across a region-to-region boundary (e.g. Northward across a zonal section) and $\delta_{ij} = -1$ if flow from $i$ to $j$ implies 'negative' transport (e.g. southward) and $\delta_{ij} = 0$ if water masses $i$ and $j$ are not in adjacent regions. We only consider region-to-region boundaries where the total mass transport is zero.

L350) I feel that this is an important point and could be highlighted more. Essentially, this technique could help provide a better estimate of the net radiative imbalance in the climate system, which is hard to do!

Thank you. We have tried to emphasise it in the entire paper.

Fig 7) Are the dots at the boundaries of regions? Where are they coming from?

We have added:

(dots, located at the boundary between adjacent regions)

Is it possible to be more continuous?

There may be ways to estimate an implied heat flux at all latitudes based on the remapped T, S and fluxes, but for simplicity we are using Equations (19) and (20), which only work for region boundaries.

 Also, this is case one and two, right?

No, but we have added the following to the caption:

(We have omitted the same figure for Case 2 since the solution is indistinguishable.)

L361) "polar regions" – refer to Fig. 9 here.

Done

Fig 8) Mention which case is shown here.

We have tried to make it clearer which cases are considered in each figure.

L365) Do you understand why the freshwater is more successful as a non-uniform pattern? Does the fact that the heat fluxes are minimized as a uniform pattern mean that the optimization problem might be set up imperfectly? Not the you need to do redo it, but it would help if you could mention of why this is so, if you have intuition about it.

We do. We have added

over relatively saline regions such as the sub-tropical oceans and the majority of the Atlantic Basin. This is likely because greenhouse forcing in ACCESS is consistent with the 'wet gets wetter, dry gets drier' paradigm (Durack et al., 2012; Skliris et al., 2016) and the consequent changes in salinity can not be affected by mixing, which can only make fresh water salty and salty water fresh (Zika et al., 2015b).

L370) I'm not sure what "increases the cost function" means. Do you mean that it "is not a minimum of the cost function?"

Changed to:

such additions are penalised since the inverse method searches for the solution with the smallest root mean squared $Q^{adjust}$

L390-400) I'm curious about other dynamical ways to constrain the problem. Would it be possible to include a feature of the weights that discounts net volume transport across the strong meridional buoyancy gradients (for instance, incorporating the tendency for advection to be along-isopycnal)? This doesn't need to be discussed in the paper, I'm just curious. In the text, however, I again suggest expanding on why the assumptions regarding the prior knowledge of g and Q, were made here from a dynamical standpoint.

Yes this would certainly be possible. One could add extra terms in the cost function and also add hard constraints. Some idea of the geographical location and gradients and not just T-S properties is likely needed as two water masses which have the same density may be very far apart geographically, while tow waters that have different densities may be very tightly stacked vertically.

Due to the very large range of possible ways to take this method we have tried our best to keep it simple in this first, proof of concept, study. We think we have been sufficiently up front about that, particularly in the discussion section.

Fig 10) Could a panel be added with the truth? This is not required if it's really complicated... (and truth would only include the total fluxes, content change, and transport, I think). But would be nice to compare to.

We have commented that

> For Case 1, the terms are indistinguishable from their `true' values in the ACCESS model.

L417) I would again suggest highlighting the point here that "this implies that the method, leveraged with observations, might help to refine observationally-based estimate of the net heat flux imbalance in the climate system." Or something....

Thank you very much. We have added:

> This implies that the method, leveraged with observations, can help to refine observationally-based estimates of the net heat and fresh water flux imbalance in the climate system.

We have completed each of the edits below.

Grammatical edits:
L31) Comma before "which"
L54) in space "and time"
L88) add quotations around "conservative"
L131) comma before "which"L175) comma after "implausible"
L264) comma before "which"
L265) I would suggest replacing "volume" with "mass" since you used mass before..
L360) Remove second "of"
L361) "Adjusted" (?) [Changed to: "…pattern of adjustments to the heat flux…"]

Review of "An optimal transformation method for inferring ocean tracer sources and sinks" by Zika & Sohail for EGUsphere.

The paper presents a new approach (Optimal Transformation Method), rooted in water mass transformation methods, to infer changes in tracer distributions in the ocean interior as a result of ocean transport (circulation and mixing) and tracer sources/sinks. The novelty of this method is that it allows to separate the effect of air-sea fluxes, which often have biases, and mixing; this separation is not usually allowed by other inverse techniques. Also, the OTM method is not based on a steady state ocean circulation assumption, hence allowing to investigate changes in the ocean circulation.

The authors present an application of this new framework to a historical numerical model, after discussing the framework details with idealised case scenarios. This new framework is an interesting new approach, complimentary to other existing methods. The paper is overall very well written and some of the technical aspects of the methodology are clearly explained. I think this manuscript fits well in EGUsphere. Before publication, I think there are some aspects of the paper that need clarification. These are overall minor revisions, discussed below.

Thank you for the encouraging comments. We have addressed all the suggestions below.

Comments:

Line 48: More than in GF, it seems to me the method is rooted in transport matrix and water mass theory..?

We agree that GF in particular is a bit too narrow as a basis and have simply changed the sentence to

> The method we propose is rooted in both ocean transport and water mass theory, both of which we will review briefly in the context of state estimation.

Line 118 (and following discussion at lines 123-127): Perhaps it might be worth to introduce a definition of a water mass? In the usual definition, which might not apply here, a water mass is defined as a "body of water with common formation history", or a "body of water whose conservative properties are set by a single, identifiable process (and altered only by mixing)". The conservative properties defining a water mass are most often set at the surface (some non-conservative properties can be acquired in the interior, e.g. an oxygen minimum, but most often that is not the case). Hence, why we usually describe properties in the interior as a linear combination of surface properties. My understanding is that in the OTM approach, the definition of a "water mass" is looser than the convention (e.g. line

118: using the definitions above, the mix of two known water masses is not a new, separate water mass), so it might be worth stating this difference from a conventional definition.

This is a good point and we have addressed it by adding the following sentence at the start of that paragraph:

> A water mass is typically defined as a body of water with distinct thermodynamic and/or chemical properties.

And then, after discussing the effect of sources and sinks and mixing on properties, we end the paragraph with:

> Because sources and sinks of properties and mixing are typically far larger near the sea surface than the deep ocean, the properties of water masses are often thought to indicate a common formation history.

Line 134: The reference to EMD is a bit confusing. Maybe I got it wrong, but my understanding is that Qi,j is the distance in tracer space between the early and late water masses due to sources/sinks. If that's the case, it might be beneficial to write that explicitly in the definition of Qi,j at line 134, so that the following statement might become less confusing. Or rephrase/expand on the EMD reference (also because you are not using the EMD in the OTM, right?)

We have removed our lengthy description of EMD as it was causing more confusion than clarity.

Line 137: I think clarifying the point above about Qi,j definition would help to better understanding Eq.5. I was initially confused about gi,j acted on Qi,j.

We recognise that this was not sufficiently clear. We have attempted to clarify the above point with the following text modifications:

> The fraction of our $i$th early water mass which is transported to the $j$th late water mass can be subjected to a source or sink of tracer on its route from one to the other. We represent this source as an implied change in tracer concentrations $\mathbf{Q}_{ij}$ along the Lagrangian path taken by the fraction of water $g_{ij}$. That is, the fraction of water ($g_{ij}$) that leaves the early water mass $i$ with tracer concentration $\mathbf{C}_{0,j}$ can be thought of as having been changed to concentration $\mathbf{C}_{0,j} + \mathbf{Q}_{ij}$ by the time it arrives at late water mass $j$.
>
> The late water mass $j$ is formed from the mixture of all the fractions of $g_{ij}$ modified along their respective paths such that its tracer concentration is
>
> $$\mathbf{C}_{1,j} = \sum_{i=1}^{N} g_{ij} \left( \mathbf{C}_{0,i} + \mathbf{Q}_{ij} \right). \tag{4}$$
>
> This provides a complete description of water mass change: the late water masses ($\mathbf{C}_{1,j}$) are formed as the linear combination of fractions ($g_{ij}$) of the early water masses ($\mathbf{C}_{0,i}$), each modified on route by sources and sinks ($\mathbf{Q}_{ij}$).

Line 148: What is the reasoning here? The previous statement says that the confidence in Qi,j is low, hence the confidence in the prior is low, correct? Why should the solution assume that Qadjust is small?

We have added the following text to clarify our reasoning and emphasise that these choices are not hard wired into the method itself but reflect what we think is a reasonable proof of concept:

In practise, we do not know any of the 4 terms in (4) with certainty for any tracers in the ocean. We can, however, frame (4) as an inverse problem, and adjust the terms within it to find solutions under certain constraints. Many different strategies could be employed depending on the confidence of the user in the different terms and constraints. We will develop and implement one approach we consider relevant to understanding recent multi-decadal changes in ocean temperature and salinity.

For heat and salt, we consider there to be relatively good confidence in observational estimates of $C_{1,j}$ and $C_{0,i}$, poorer confidence in estimates of $Q_{ij}$ and poor to no confidence in estimates of $g_{ij}$. The concentrations $C_{1,j}$ and $C_{0,i}$ can be derived from ocean temperature and salinity analyses (e.g. (Good et al., 2013)). These come with substantial uncertainties (Cheng et al., 2022; Stammer et al., 2021), but have the benefit of essentially being mappings of directly observed quantities. The source/sink term $Q_{ij}$ can be inferred from air-sea flux products but these come with larger uncertainties. For example, heat content changes derived from temperature analyses vary by order $0.1 \text{W/m}^2$ (e.g. $0.05 \text{W/m}^2$ for 1958-2019, Cheng et al., 2022) while those derived from accumulated air-sea heat fluxes typically have biases of order $1 \text{W/m}^2$ (e.g. $4 \text{W/m}^2$ for 1993-2009, Valdivieso et al., 2017). Finally, we know of no direct way of deriving $g_{ij}$ from observations. Indeed, $g_{ij}$ could be derived from a data constrained numerical model, but that would imply it is indirectly derived from the same data used for $C_{1,j}$, $C_{0,i}$ and $Q_{ij}$. We thus consider it reasonable to frame an inverse problem where $C_{1,j}$ and $C_{0,i}$ are considered 'known', priors for $Q_{ij}$ are provided and $g_{ij}$ is merely constrained to obey laws of physics.

Line 150: I might have missed it, but why is the cost function in eq. 7 called called "non-mixing cost"?

We have now simply changed it to "[Cost]"

Also, it was not until I read the Results section that it became clear that the steps are to (i) solve for gi,j in (7) and (ii) then calculate Qadjust in (8). I would suggest to state more clearly here.

Indeed. We have added:

To recapitulate, we have described a method where we find the optimal transport matrix $g_{ij}$ using (6), and then, from this, we find the adjustment required to tracer sources and sinks using (7). We call this an Optimal Transformation Method (OTM) since we are looking for the optimal way in which the waters can be transformed to describe the evolving ocean state given our physical constraints.

Fig 4: I am a bit confused by this figure. If I understand correctly, first the ocean is split in 16 T-S groups of equal global volume, and fig. 4 shows the volume of each of this groups in the 9 geographical regions considered. So, if we were to sum up the volumes across all nine regions per each water mass, we should retrieve the same volume?

I might be reading this wrong, but I do not see this in Fig.4. Take for example the water mass defined by T[-2:5] and S[30:34.5] approx (bottom left box). This water mass has a relatively low volume compared to other water masses in almost all regions (but N. Pac, perhaps). It overall seems to be orders of magnitude lower than

the volume of the water mass defined by T[-2:4] and S[34.5:35.5] approx. Again, I might be reading this wrong, so some clarification would be appreciated.

To avoid too much emphasis on the abyss, we used 0-2000m to define our bin edges. We use the full volume in the analysis shown, just not to define the bin edges (Section 3.3: Implementation of the inverse model).

Also: (1): it would be useful to add these boxes in Fig.3 and use the same x and y intervals and spacing, if possible;

Done

(2) only 14 of the water masses are visible in Fig.4, maybe change the axis to improve visualisation?

The changes to Fig. 3 now make it much easier for the reader to see where the water mass boundaries are.

Eq. 9: Ai is not the outcrop area at the early stage (right?), which is would one would most likely assume. Is it the outcrop area while transitioning between early and late periods? Some clarification would be useful.

For clarity, we have moved the definition of the average outcrop area to after the definition of the prior sources and sinks and added amended the text with the following:

*bias* is a bias we will introduce in some cases to see what effect incorrect air-sea flux data has on the inverse solution. The above time integral is from the midpoint of the early period to the midpoint of the late period since it is related to the change in average water mass properties between the two periods. (Integrating from the start of the early to the end of the later period would overestimate the sources and sinks.)

In our implementation of the optimisation (13) we aim to minimise the average adjustment to the tracer sources and sinks in a per unit area sense. For this reason we calculate the average outcrop area using the same integral limits as the sources and sinks such that

$$A_i = \frac{1}{t_1 - t_0} \int_{t_0}^{t_1} \iint \Omega_i(x, y, 0, t) \, dx \, dy \, dt. \tag{16}$$

Also, should \Omega_i be \Omega_i(x,y,t) only (also in Eq. 19)?

\Omega_i is a 3D and time dependent field of 1's and 0's. For clarity we have changed \Omega_i({\bf x},t) to \Omega_i(x,y,z,t)

Line 311-312: Why don't you attribute different adjustments, but Qadjust is the same for all i?

Because having a different Qadjust for each i and j would quadratically increase the number of degrees of freedom.

Fig. 6: Perhaps change the colorbar for Qadjust, so that they are not just blank? Or remove the figure and just use the signal to noise reported (lines 328-329) to make the point that Qadjust << Qprior?

We have changed the colorbar for Qadjust in Fig 6 so the pattern of the (very small) adjustment can be seen.

Fig. 7: The number of points where transports can be inferred is limited by the number of regions selected, correct? Also, perhaps add the inferred transports for Case 2 and show that they are indistinguishable and change caption to mention both Case 1 and 2?

We now say in the caption:

> (We have omitted the same figure for Case 2 since the solution is indistinguishable.)

Line 344: Can you add a justification of why you selected 5 W/m2 for the heat flux bias, and 5 mm/day for the fresh water flux bias? Why not larger/smaller (well, I guess larger would be more interesting) biases? And why not a percentage of the signal, rather than a fixed amount? And what if the biases were not uniformly positive/negative? Maybe I missed the point, but how could a fixed Qadjust reflect a mix positive/negative biases?

A new analysis with a non-uniform bias with more practical relevance has been added as Case 3 in the manuscript.

Line 361-363: I think I am off here, comparing apple and oranges, but how does the result for the heat flux compare with the redistributed vs added heat? Can we interpret fig.9 (for the heat flux changes) as an indication that most of the ocean heat content changes are described by redistributed heat (gi,j explaining most of it), and only part of the changes are caused by added heat?

We have removed most of our discussion of added and re-distributed heat and our previous work with the EMD algorithm. It is therefore a bit tangential to go into a discussion here of how these calculations may relate to added and redistributed heat.

Minor comments:
All edits below have been done.
Line 18: Estimates (capital E)

Line 19: Remove "However"?
Line 38: Delete "[" at the end of the line
Line 70: "properties" misspelled

Line 151: Add reference to section: "where wj is a relevant weighting (see section 2.5)".

Line 173: "early" and "late" in wrong order.

Fig 3: colorbar label has kg spelled differently in the same label (Kg and kg)

In fact it is "K g / kg"

Line 291: Eq 15 (and not 7)?

Line 339: we "find" (verb missing)

Line 360: two "of"

Line 361: add reference to fig. 9. Also, maybe change the colorbar for the adjusted heat flux?